# CB-SLICE: Concept-Based Interpretable Error Slice Discovery

Yael Konforti [1]  Mateo Espinosa Zarlenga [1 2]  Elaf Almahmoud [1 3]  Mateja Jamnik [1]

## Abstract

Despite strong average-case performance, deep learning models often exhibit systematic errors on specific population groups, known as *error slices*. Identifying these groups and the root causes of their failures is critical for model debugging and bias mitigation. However, existing error Slice Discovery Methods (SDMs) typically generate explanations disconnected from the model's inference process, thus only approximating the underlying error source and may be inaccurate. We address this limitation by leveraging Concept Bottleneck Models (CBMs), whose predictions are directly dependent on human-understandable semantic concepts. Since downstream task failures in CBMs commonly arise from concept mispredictions, concept representations provide a strong candidate for error slice identification, offering fine-grained explanations directly linked to the error source. Building on this insight, we introduce *CB-SLICE*, a concept-based SDM that groups samples with shared concept prediction failures and identifies the keyword-concepts most responsible for each slice's failure-mode. Across multiple benchmarks, we show that CB-SLICE outperforms state-of-the-art methods in uncovering well-known biases while providing richer and more faithful explanations of model errors.

## 1. Introduction

Machine learning (ML) models have achieved remarkable average-case performance across a wide range of tasks (He et al., 2016; Radford et al., 2021; Rombach et al., 2022; Liu et al., 2024; Oquab et al., 2024) and are increasingly deployed in critical domains such as healthcare (Lee & Yoon, 2021), criminal justice (Dressel & Farid, 2018), and job recruitment (Van Esch et al., 2019). Despite this progress, ML models remain susceptible to biases and often exhibit systematic failures on specific subsets of data, commonly referred to as *error slices* (Eyuboglu et al., 2022). Such failures can disproportionately affect certain populations (Ntoutsi et al., 2020), amplify societal disparities (Mehrabi et al., 2021), and thus compromise deployment safety. For example, Vieira et al. (2025) show that skin lesion classifiers consistently underperform on the minority melanoma class across standard benchmarks. Daneshjou et al. (2022) further demonstrate substantial performance disparities across skin tones, with state-of-the-art models performing noticeably worse on darker skin types due to their underrepresentation in the training dataset, with direct implications for clinical reliability and quality of care.

Identifying and understanding such biases is essential for improving model robustness (Sagawa et al., 2019) and fairness (Kim et al., 2019), and for guiding effective dataset curation (Liang et al., 2022). However, error slice discovery poses two fundamental challenges: (i) identifying coherent subsets of erroneous samples that share a well-defined semantic failure mode (e.g., darker skin melanoma lesions), and (ii) explaining the underlying cause of failure in terms of human-understandable *concepts* (e.g., *"dark skin"*) on which the model relies.

To address this challenge, various error Slice Discovery Methods (SDMs) have been proposed (Eyuboglu et al., 2022; Chen et al., 2023; Jain et al., 2023; Rezaei et al., 2024). While they can recover well-known failure modes with high precision, they typically rely on auxiliary language-based models (e.g., ClipCap (Mokady et al., 2021)) to generate explanations. Consequently, the explanations are disconnected from the analysed model and its internal decision-making process, making them only indirectly related to the true error source, and thereby inaccurate and misleading (Rudin, 2019; Bordt et al., 2022). Moreover, they may inherit biases from the auxiliary model, further undermining reliability.

Here, we address this limitation by reframing error slice discovery through the lens of Concept Bottleneck Models (CBMs) (Koh et al., 2020; Espinosa Zarlenga et al., 2022; Kim et al., 2023). CBMs are interpretable deep neural

---

[1]Department of Computer Science and Technology, University of Cambridge, Cambridge, UK [2]Trinity College, University of Oxford, Oxford, UK [3]Cambridge Institute for Technology and Humanity, Cambridge, UK. Correspondence to: Yael Konforti <yk449@cam.ac.uk>.

*Proceedings of the 43rd International Conference on Machine Learning*, Seoul, South Korea. PMLR 306, 2026. Copyright 2026 by the author(s).

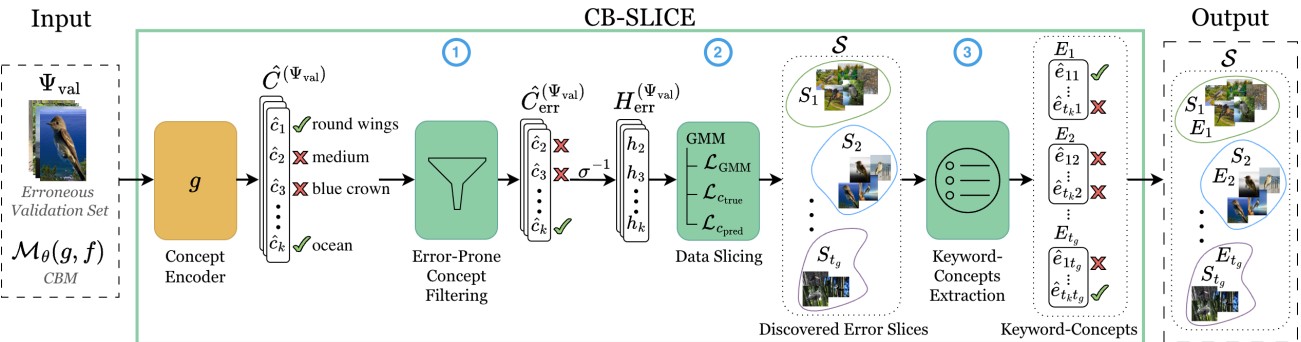

**Figure 1.** Given $\Psi_{\text{val}}$ and $\mathcal{M}_\theta(g, f)$, CB-SLICE discovers and explains systematic failure modes in three steps. (1) *Error-prone concept filtering:* the concept encoder $g$ produces concept predictions $\hat{C}^{(\Psi_{\text{val}})}$ from which error-prone concepts are selected to form $\hat{C}_{\text{err}}^{(\Psi_{\text{val}})}$. (2) *Error slice formation:* the corresponding concept logits $H_{\text{err}}^{(\Psi_{\text{val}})}$ are clustered via a GMM to obtain error slices $\mathcal{S}$. (3) *Failure mode explanation:* for each slice $S_j$, its keyword-concept predictions $E_j$ are extracted, yielding concept-level explanations.

networks (DNNs) that first predict human-understandable concepts (e.g., *"dark skin"*, *"asymmetry"*), and then predict the class label (e.g., *"Melanoma"*) based on the concept predictions. This structured prediction pipeline creates a direct and transparent link between the model's downstream task predictions and semantic concepts, making CBMs a natural foundation for faithful failure analysis.

Motivated by this observation, we introduce **CB-SLICE**[1], a concept-based SDM that integrates error slice discovery and bias explanation into a single, model-aware process by utilising the CBM architecture. Our key insight is that when downstream predictions depend on intermediate concept predictions, systematic errors must originate from these concept predictions. By leveraging CBMs' support of *concept interventions* – modifications to concept representations during inference – we can directly quantify which concepts drive downstream errors. CB-SLICE operates in three steps (Figure 1): (1) identify *error-prone concepts* that contribute most to downstream errors; (2) cluster erroneous validation samples in the concepts' logit space, restricted to the *error-prone concept* set, to form slices that capture shared concept-level error patterns; and (3) explain each slice by identifying the concepts most responsible for its formation. Finally, because analysing even a moderate number of slices can be time-consuming and impractical, we propose a slice prioritisation strategy that highlights the most *informative* slices to enable scalable and focused model debugging. Our contributions are summarised as follows:

1. **CB-SLICE:** a novel SDM that links discovered explanatory keywords directly to the model's internal decision logic via CBMs.

2. **Error-prone concept identification:** a principled method to identify error-prone concepts for both slice formation and failure mode characterisation.

3. **Slice prioritisation:** a strategy that guides practitioners towards the most informative slices for targeted analysis and downstream mitigation.

4. **Empirical results:** consistent improvements over state-of-the-art baselines across benchmarks in recovering known failure modes, while providing informative and faithful explanations of the error source.

## 2. Background and Related Work

**Concept-Bottleneck Models** CBMs (Koh et al., 2020) provide a principled architecture for training interpretable DNNs based on human-understandable concepts. In this framework, we consider a classification task where $\mathbf{x} \in \mathcal{X}$ denotes an input sample, $y \in \mathcal{Y}$ a task label, and $\mathbf{c} \in \{0, 1\}^k$ a binary concept annotation vector consisting of $k$ concepts. These concepts are given as training labels and can be provided by experts (Koh et al., 2020) or discovered (Oikarinen et al., 2023; Yang et al., 2023). A CBM $\mathcal{M}_\theta$, parametrised by $\theta$, consists of two components: (1) a concept encoder $g : \mathcal{X} \mapsto [0, 1]^k$, which maps the input samples to concept predictions, and (2) a label predictor $f : \mathcal{C} \mapsto \mathcal{Y}$, which maps concept predictions to a downstream task space. Training can be done in three ways: (a) *independently*, where $g$ and $f$ are trained separately; (b) *sequentially*, where $g$ is trained first and then $f$ is trained using $g$'s outputs; and (c) *jointly*, where both $g$ and $f$ are trained simultaneously.

In addition, CBMs support *concept interventions*, where concept predictions can be modified at inference time to alter downstream outputs. This enables us to directly quantify

---

[1]Our code can be found at https://github.com/yaelkon/CB-SLICE.

which concepts affect the final predictions and is therefore valuable for pinpointing those that drive errors (Section 3).

**CBMs for Failure Modes Discovery**   This work studies the utility of CBMs in discovering and explaining failure modes for dataset debugging. Accordingly, it is related to prior works on bias identification and mitigation in CBMs. For example, Bordt et al. (2022) suggest mitigating spurious concepts by pruning their corresponding classifier weights, while Enouen et al. (2025) suggest doing so by removing them from the concept bank and then retraining. However, identifying such concepts requires substantial human supervision. To reduce this burden, Kim et al. (2024a) leverage vision-language models (Liu et al., 2024) to automatically curate concept banks without spurious concepts. Penaloza et al. (2025) address concept mislabeling by incorporating posterior concept density into the learning process. In contrast, we aim to provide a debugging tool for systematically uncovering all failure modes rather than focusing on mitigating specific types of bias.

**Error Slice Discovery and Failure Mode Explanation** Prior SDMs typically leverage the latent embedding space of trained models. For example, GEORGE (Sohoni et al., 2020) uses a 2D projection of the model's embeddings and clusters samples within each class. Spotlight (d'Eon et al., 2022) identifies dense, high-loss regions for which the model underperforms, while MCSD (Yu et al., 2025) further adds a compactness constraint to the slicing objective. These methods identify error slices but do not explain their underlying causes, limiting their usefulness for debugging.

To address this gap, recent works use language models to explain failure modes. For example, Domino (Eyuboglu et al., 2022) discovers error slices in CLIP's joint image–text embedding space (Radford et al., 2021) using an error-aware objective that groups samples with similar mispredictions, and then retrieves textual descriptions aligned with each slice. Similarly, Jain et al. (2023) use CLIP to identify class-specific failure directions, while FACTS (Yenamandra et al., 2023) amplifies model biases before discovering slices. Other SDMs suggest first generating interpretable concepts characterising the dataset and then identifying failure modes by evaluating the model performance across concept combinations (Chen et al., 2023; 2025; Rezaei et al., 2024). However, these methods rely on auxiliary models disconnected from the analysed model's internal logic, which limits the explanation fidelity and risks being inaccurate (Rudin, 2019; Bordt et al., 2022) or biased (Gallegos et al., 2024). We address this challenge by integrating slice discovery and explanation into the concept latent space of CBMs, thereby directly tying them to model decisions.

## 3. CB-SLICE

In this section, we introduce CB-SLICE, a concept-based SDM that integrates failure mode explanations into the slice discovery process by leveraging the CBM architecture.

To understand how CB-SLICE operates, assume we have access to a CBM $\mathcal{M}_\theta = (g, f)$ trained on a dataset $\mathcal{D}_{\text{train}}$ and that $\Psi_{\text{val}} \subseteq \mathcal{D}_{\text{val}}$ is a set of samples $(\mathbf{x}, \mathbf{c}, y)$ in a hold-out validation set $\mathcal{D}_{\text{val}}$ where $\mathcal{M}_\theta$ mispredicts the downstream task label $y$. CB-SLICE (Figure 1) then proceeds in three steps: (1) filtering error-prone concepts, (2) forming error slices, and (3) explaining the failure modes captured by each discovered slice. Below, we describe each step in detail.

### 3.1. Step 1 - Error-Prone Concepts Filtering

Given $\mathcal{M}_\theta$ and $\Psi_{\text{val}}$, we first aim to identify a subset of *error-prone concepts* that are most likely responsible for the observed misclassifications in $\Psi_{\text{val}}$, to ensure that the discovered error slices are driven by concepts that induce downstream errors. To this end, we introduce an *error-prone concept filtering* algorithm that identifies a set of concepts that contribute most to the model's failures. This is done by leveraging the Expected Change in Target Prediction (ECTP) score (Shin et al., 2023), which quantifies the expected change in the downstream task distribution when one intervenes on a specific concept.

Specifically, we first generate the concept predictions for the erroneous validation set $\hat{C}^{(\Psi_{\text{val}})}$, as illustrated in Figure 1. Then, for each predicted concept indexed $i$ within this set $\{\hat{c}_i\}_{i=1}^k \in \hat{C}^{(\Psi_{\text{val}})}$, we compute its ECTP score $\mathrm{T}_i(\hat{\mathbf{c}})$ as

$$\mathrm{T}_i(\hat{\mathbf{c}}) = (1 - \hat{c}_i) D_{\text{KL}}(\hat{y}_{\hat{c}_i=0} \parallel \hat{y}) + \hat{c}_i D_{\text{KL}}(\hat{y}_{\hat{c}_i=1} \parallel \hat{y}), \quad (1)$$

where $\hat{c}_i \in [0, 1]$ is the probability of observing the $i$-th concept produced by the concept encoder $g$, $\hat{y}$ is the original predicted class distribution, $\hat{y}_{\hat{c}_i=v}$ is the predicted class distribution obtained by intervening on concept $i$ and setting it to $v \in \{0, 1\}$, and $D_{\text{KL}}$ is the Kullback-Leibler divergence (Csiszár, 1975).

As failures of different output targets may originate from diverse concepts, we compute the class-wise ECTP score to mitigate class imbalance effects:

$$\overline{\mathrm{T}}_i^{(l)} = \frac{1}{\mid \Psi_{\text{val}}^{(l)} \mid} \sum_{\mathbf{x} \in \Psi_{\text{val}}^{(l)}} \mathrm{T}_i(g(\mathbf{x})), \quad (2)$$

where $\hat{\mathbf{c}} = g(\mathbf{x})$ and $\Psi_{\text{val}}^{(l)} \subseteq \Psi_{\text{val}}$ denote the set of erroneous validation samples with ground-truth label $y = l$. Then, for each class $l$, we select the top $t_e$ concepts with the highest $\overline{\mathrm{T}}_i^{(l)}$ values as the *class error-prone concepts*, where $t_e$ is a user-selected hyperparameter. The set specified by these concepts forms the *error-prone concept set* $C_{\text{err}}$, where $|C_{\text{err}}| = k_{\text{err}}$ and $k_{\text{err}} \leq k$. We empirically validate this step

in Appendix A.1, showing that restricting slice formation to error-prone concepts substantially improves CB-SLICE performance[2] compared to using the full concept set.

## 3.2. Step 2 - Forming Error Slices

After identifying the error-prone concept set $C_{\text{err}}$, our goal is to form coherent error slices that capture systematic concept-level failure patterns. In CBMs, downstream task errors typically arise from concept mispredictions. Therefore, concept representations provide a natural and fine-grained space for isolating failure modes. Let $\hat{C}_{\text{err}}^{(\Psi_{\text{val}})}$ denote the collection of concept predictions of samples in $\Psi_{\text{val}}$, restricted to $C_{\text{err}}$. Instead of forming slices in the probability space, CB-SLICE uses the corresponding concept-logit representations $H_{\text{err}}^{(\Psi_{\text{val}})} = \sigma^{-1}\big(\hat{C}_{\text{err}}^{(\Psi_{\text{val}})}\big)$, where $\sigma^{-1}$ is the logit function. Concept logits encode the model's confidence in a concept's presence, and have been shown to be well approximated by Gaussian distributions (Vandenhirtz et al., 2024). Thus, $H_{\text{err}}^{(\Psi_{\text{val}})}$ provides a semantically meaningful space for identifying error patterns and enables statistical modelling.

We form slices by clustering $H_{\text{err}}^{(\Psi_{\text{val}})}$ using a Gaussian Mixture Model (GMM). Specifically, we learn $t_g$ slices $\mathcal{S} = \{S_1, \ldots, S_{t_g}\}$, where $t_g$ is a tuned hyperparameter. Each slice $S_j$ is parametrised by $\Omega_j = \big(\mu_j, \Sigma_j, \alpha_j\big)$, with mean $\mu_j \in \mathbb{R}^{k_{\text{err}}}$, covariance $\Sigma_j \in \mathbb{S}_{++}^{k_{\text{err}} \times k_{\text{err}}}$ (the set of $k_{\text{err}} \times k_{\text{err}}$ symmetric positive definite matrices), and prior weights $\alpha_j \in \mathbb{R}_+$ satisfying $\sum_{j=1}^{t_g} \alpha_j = 1$. For computational efficiency, we learn diagonal covariance matrices. The GMM is implemented as a differentiable layer (Konforti et al., 2022) and trained via stochastic gradient descent.

To encourage semantic coherence, we minimise the negative log-likelihood over $H_{\text{err}}^{(\Psi_{\text{val}})}$:

$$\mathcal{L}_{\text{GMM}} = -\sum_{\mathbf{h} \in H_{\text{err}}^{(\Psi_{\text{val}})}} \log\left(\sum_{j=1}^{t_g} \alpha_j \mathcal{N}(\mathbf{h} \mid \mu_j, \Sigma_j)\right) \quad (3)$$

Beyond semantic similarity, we wish to form slices whose members share the same concept-level error patterns. Inspired by Eyuboglu et al. (2022), we introduce two auxiliary linear classifiers: $z_c(\cdot)$ predicts ground-truth concept labels, and $z_{\hat{c}}(\cdot)$ predicts concept predictions based on the probability of sample $\mathbf{x}$ belonging to slice $S_j$:

$$P(S_j \mid \mathbf{x}) = \frac{P(\mathbf{x}, S_j)}{P(\mathbf{x})} = \frac{\alpha_j \mathcal{N}(\mathbf{h} \mid \mu_j, \Sigma_j)}{\sum_{n=1}^{t_g} \alpha_n \mathcal{N}(\mathbf{h} \mid \mu_n, \Sigma_n)}. \quad (4)$$

For each $\mathbf{x} \in \Psi_{\text{val}}$, let $\mathbf{c}_{\text{err}}$ and $\hat{\mathbf{c}}_{\text{err}}$ denote its ground-truth and predicted concept vectors, restricted to $C_{err}$. Then, the

auxiliary losses are:

$$\begin{aligned}
\mathcal{L}_{c_{\text{true}}} &= \sum_{\mathbf{x} \in \Psi_{\text{val}}} \text{BCE}\Big(z_c(\mathbf{r}), \mathbf{c}_{\text{err}}\Big), \\
\mathcal{L}_{c_{\text{pred}}} &= \sum_{\mathbf{x} \in \Psi_{\text{val}}} \text{BCE}\Big(z_{\hat{c}}(\mathbf{r}), \hat{\mathbf{c}}_{\text{err}}\Big),
\end{aligned} \quad (5)$$

where $\mathbf{r} = [P(S_j \mid \mathbf{x})]_{j=1}^{t_g}$ and BCE is the binary cross-entropy function. Intuitively, $\mathcal{L}_{c_{\text{true}}}$ encourages slices whose membership is predictive of ground-truth concept labels, pushing members of the same slice to share the same true concept values. $\mathcal{L}_{c_{\text{pred}}}$ applies the same idea to concept predictions, ensuring that slice members also share the same concept prediction values. As a result, each slice captures a coherent concept-level error pattern. The overall slice formation objective is:

$$\mathcal{L} = \mathcal{L}_{\text{GMM}} + \lambda(\mathcal{L}_{c_{\text{true}}} + \mathcal{L}_{c_{\text{pred}}}) \quad (6)$$

where $\lambda$ is a hyperparameter that balances semantic coherence against alignment in concept-level error patterns. The contribution of each loss term is empirically validated in Appendix A.2, where we show that combining the GMM objective with the two auxiliary classification losses yields the highest and most consistent performance[2]. Additionally, we examine the choice of GMM over a linear alternative in Appendix A.3, using $\mathcal{L}_{c_{\text{true}}}$ and $\mathcal{L}_{c_{\text{pred}}}$ accuracies as performance measures, and show that GMM-based slicing consistently outperforms linear clustering.

## 3.3. Step 3 - Explaining Slice Failure Modes

Finally, for each discovered error slice $S_j \in \mathcal{S}$, we wish to explain the failure mode it captures by identifying the concepts that most characterise its formation. To this end, we extend the ECTP score (Eq. 1) to quantify the effect of concept interventions on slice assignment rather than task prediction. Concepts whose intervention induces the largest change in slice membership are set to be the slice's explanatory *keyword-concepts*.

Concretely, for each $\mathbf{x} \in S_j$ and concept $i$, we measure how intervening on the predicted concept value $\hat{c}_i$ alters the posterior slice assignment probability $P(S_j \mid \mathbf{x})$. We capture this via the *Expected Change in Slice Assignment* (ECSA):

$$\text{ECSA}_i(\mathbf{x}) = \mathbb{E}_{v \sim \text{Bern}(\hat{c}_i)}\Big[D_{\text{KL}}\Big(P(S_j \mid \mathbf{x}, \hat{c}_i = v) \parallel P(S_j \mid \mathbf{x})\Big)\Big], \quad (7)$$

where $P(S_j \mid \mathbf{x}, \hat{c}_i = v)$ denotes the likelihood of sample $\mathbf{x}$ belonging to slice $S_j$ after intervening on concept $i$ and setting its predicted value to $v \in \{0, 1\}$. We then average the ECSA score of each concept across all members of $S_j$ and select the top $t_k$ concepts as the slice's *keyword-concepts* $E_j = \{\hat{e}_{1j}, \ldots, \hat{e}_{t_k j}\}$, where $t_k$ is a user-selected hyperparameter. The value of each selected concept (present or

---

[2]Measured using the evaluation metrics defined in Section 5.3.

absent) is determined by the weighted average of its predictions across slice members, using $P(S_j \mid \mathbf{x})$ as weights. The resulting keyword-concepts provide a compact, model-grounded explanation of the slice's failure mode.

## 4. Slice Prioritisation

Although many error slices may be discovered, analysing them all is time-consuming and often impractical. Moreover, some slices reflect noisy failures rather than systematic error modes, limiting their debugging value. We therefore introduce a slice prioritisation strategy that ranks slices by informativeness, defined by both high *misprediction coherence*, the consistency of downstream task mispredictions within a slice, and *semantic compactness*, the degree to which the slice is concentrated in embedding space.

**Misprediction Coherence**   Meaningful error slices should exhibit agreement among their members with respect to task misprediction. Let $N_j^{\text{eff}} = \sum_{\mathbf{x} \in S_j} P(S_j \mid \mathbf{x})$ denote the effective size of slice $S_j$. Then, we define the $j$-th slice *misprediction coherence* (MC) as a normalised score (recall that $\hat{y}$ denotes the CBM's predicted class label):

$$
\begin{aligned}
\text{MC}_j &= 1 - \frac{H(\mathbf{p}_j)}{\log L} = 1 - \frac{-\sum_{l=1}^{L} p_j^{(l)} \log p_j^{(l)}}{\log L}, \\
p_j^{(l)} &= \frac{\sum_{\mathbf{x} \in S_j} P(S_j \mid \mathbf{x}) \, \mathbb{I}[\hat{y} = l]}{N_j^{\text{eff}}}, \qquad l \in \{1, \dots, L\},
\end{aligned}
\tag{8}
$$

where $H(\mathbf{p}_j)$ is the entropy of the misprediction label distribution within the slice, and $L$ is the number of classes. Higher values indicate stronger agreement on the mispredicted outcome.

**Semantic Compactness**   Informative slices should be semantically coherent. Let $\mathbf{m}_j \in \mathbb{R}^{k_{\text{err}}}$ denote the centroid of slice $S_j$ in concept-logit space. We define semantic compactness as

$$
\text{SC}_j = \frac{1}{N_j^{\text{eff}}} \sum_{\mathbf{x} \in S_j} P(S_j \mid \mathbf{x}) \cos(\mathbf{h}, \mathbf{m}_j),
\tag{9}
$$

which measures the average cosine similarity of slice members to their centroid.

**Slice Informativeness Score (SI)**   We combine the two measurements into a bounded score $\text{SI}_j \in [0, 1]$:

$$
\text{SI}_j = \rho \frac{1}{2} \Big( \text{MC}_j + \frac{1 + \text{SC}_j}{2} \Big),
\tag{10}
$$

where $\rho = 1 - \exp{-\frac{N_j^{\text{eff}}}{\tau}}$ is a penalty factor that downweights very small slices. The hyperparameter $\tau$ controls this effect. Slices with high SI scores exhibit both consistent failure behaviour and strong semantic coherence, and are therefore prioritised for analysis.

## 5. Experiments

**Research Questions**   We explore the following questions:

**(Q1)** How effective is CB-SLICE in recovering the full set of ground-truth error slice groups?

**(Q2)** How well does each discovered slice align with a single failure mode?

**(Q3)** Do CB-SLICE's keyword-concepts faithfully explain the underlying causes of the model's errors?

**(Q4)** Can CB-SLICE provide a principled way for selecting the number of discovered slices?

### 5.1. Datasets

We evaluate methods on the following datasets containing known error slices (see Appendix B for details):

**Waterbirds** (Sagawa et al., 2019) is a bird classification task (*Landbirds* vs. *Waterbirds*) where the image background (*land* vs. *water*) is spuriously correlated with the class, inducing two ground-truth error slices: *Landbirds-on-water* backgrounds and *Waterbirds-on-land* backgrounds. We use 112 bird-part concept annotations from the CUB-200-2011 dataset (Wah et al., 2011), preprocessed as in (Koh et al., 2020), in addition to the background attributes. In Waterbird's official validation set, the aforementioned spurious correlation does not hold. To keep privileged information about ground-truth error slices from leaking during model selection, we follow (Espinosa Zarlenga et al., 2025b) and resample the validation from the training set.

**CelebA** (Liu et al., 2015) is a face recognition task consisting of 40 attributes for each image. We use the sex attribute (i.e., *Male* vs. *not Male* as *Female*) as the task label [3] and the remaining 39 attributes as concepts. Following Nam et al. (2020), we consider two ground-truth error slices: *Males* with *"blonde hair"* and *Females* with *"heavy makeup"*.

**MetaShift** (Liang & Zou, 2022) is a natural dataset that consists of images of *Cats* and *Dogs*, spuriously correlated with the *"indoor"* and *"outdoor"* attributes, respectively. Thus, it induces two ground-truth error slices: *Cats "outdoors"* and *Dogs "indoors"*. In addition to the spurious attributes, we generate task-aware concept annotations using a label-free pipeline (Oikarinen et al., 2023).

**MNIST-Sum** is an MNIST-based (LeCun et al., 1998) dataset where each sample consists of two greyscale digits ($[0, 3]$) concatenated side by side, and the label corresponds to their sum (i.e., $y \in \{0, \dots, 6\}$). We define concept annotations as an 8-dimensional binary encoding of the two digits. To simulate biases, we (i) colour $90\%$ of samples con-

---

[3]We do not endorse binary gender classifiers and recognise the sex-gender distinction (Walker & Cook, 1998). We use CelebA as a testbed due to lack of standard benchmarks with known biases.

*Table 1.* Precision@10 and MGF, reported as mean $\pm$ std over five seeds. Best results per task, and those not statistically significantly different, are **bolded** and underlined. Methods are evaluated on vanilla DNN and sequentially/jointly trained CBM, except CB-SLICE, which applies only to CBMs. CB-SLICE consistently outperforms baselines across benchmarks on both metrics.

| DATASET | MODEL | DOMINO | GEORGE | HIBUG2 | SPOTLIGHT | K-MEANS | CB-SLICE (OURS) |
|---|---|---|---|---|---|---|---|
| Waterbirds | Vanilla DNN | $\mathbf{0.53}_{\pm.09}$ / $0.05_{\pm.0}$ | $0.34_{\pm.04}$ / $0.15_{\pm.01}$ | $0.25_{\pm.0}$ / $\mathbf{0.25}_{\pm.0}$ | $0.05_{\pm.0}$ / $0.05_{\pm.0}$ | $0.05_{\pm.0}$ / $0.05_{\pm.0}$ | – |
|  | CBM + Seq | $\underline{0.72}_{\pm.03}$ / $0.17_{\pm.01}$ | $0.22_{\pm.1}$ / $0.13_{\pm.07}$ | $0.2_{\pm.0}$ / $\underline{0.25}_{\pm.0}$ | $0.05_{\pm.0}$ / $0.04_{\pm.0}$ | $0.11_{\pm.04}$ / $0.03_{\pm.0}$ | $\mathbf{0.78}_{\pm.07}$ / $\mathbf{0.7}_{\pm0.05}$ |
|  | CBM + Joint | $0.62_{\pm.04}$ / $0.24_{\pm.17}$ | $0.18_{\pm.13}$ / $0.08_{\pm.03}$ | $0.25_{\pm.0}$ / $0.25_{\pm.0}$ | $0.0_{\pm.0}$ / $0.04_{\pm.0}$ | $0.09_{\pm.04}$ / $0.03_{\pm.0}$ | $\underline{0.83}_{\pm.03}$ / $\underline{0.76}_{\pm.04}$ |
| CelebA | Vanilla DNN | $\mathbf{0.61}_{\pm.12}$ / $0.49_{\pm.02}$ | $\mathbf{0.55}_{\pm.06}$ / $0.2_{\pm.0}$ | $\mathbf{0.6}_{\pm.0}$ / $\mathbf{0.58}_{\pm.0}$ | $0.25_{\pm.0}$ / $0.1_{\pm.0}$ | $0.2_{\pm.0}$ / $0.2_{\pm.0}$ | – |
|  | CBM + Seq | $\underline{0.63}_{\pm.06}$ / $0.45_{\pm.02}$ | $\underline{0.49}_{\pm.04}$ / $0.17_{\pm.01}$ | $\underline{0.55}_{\pm.0}$ / $\underline{0.51}_{\pm.0}$ | $0.0_{\pm.0}$ / $0.09_{\pm.0}$ | $0.17_{\pm.03}$ / $0.13_{\pm.0}$ | $\mathbf{0.92}_{\pm.04}$ / $\mathbf{0.66}_{\pm.07}$ |
|  | CBM + Joint | $0.64_{\pm.06}$ / $0.53_{\pm.05}$ | $0.41_{\pm.07}$ / $0.15_{\pm.0}$ | $0.4_{\pm.0}$ / $0.39_{\pm.0}$ | $0.1_{\pm.0}$ / $0.09_{\pm.0}$ | $0.12_{\pm.04}$ / $0.13_{\pm.0}$ | $\underline{0.84}_{\pm.08}$ / $\underline{0.66}_{\pm0.08}$ |
| MetaShift | Vanilla DNN | $\mathbf{0.75}_{\pm.03}$ / $\mathbf{0.58}_{\pm.07}$ | $0.64_{\pm.05}$ / $0.26_{\pm.0}$ | $0.4_{\pm.0}$ / $0.34_{\pm.0}$ | $0.07_{\pm.03}$ / $0.11_{\pm.0}$ | $0.12_{\pm.03}$ / $0.13_{\pm.01}$ | – |
|  | CBM + Seq | $0.7_{\pm.03}$ / $0.51_{\pm.06}$ | $\mathbf{0.83}_{\pm.04}$ / $0.69_{\pm.0}$ | $0.60_{\pm.0}$ / $0.49_{\pm.0}$ | $0.2_{\pm.0}$ / $0.1_{\pm.0}$ | $0.21_{\pm.04}$ / $0.16_{\pm.03}$ | $\underline{0.82}_{\pm.03}$ / $\underline{0.76}_{\pm.04}$ |
|  | CBM + Joint | $0.86_{\pm.06}$ / $0.4_{\pm.05}$ | $\underline{0.84}_{\pm.02}$ / $0.72_{\pm.0}$ | $0.5_{\pm.0}$ / $0.44_{\pm.0}$ | $0.13_{\pm.03}$ / $0.11_{\pm.0}$ | $0.15_{\pm.03}$ / $0.16_{\pm.05}$ | $\underline{0.91}_{\pm.04}$ / $\underline{0.86}_{\pm.07}$ |
| MNIST-Sum | Vanilla DNN | $\mathbf{0.44}_{\pm.1}$ / $0.14_{\pm.13}$ | $\mathbf{0.33}_{\pm.12}$ / $0.23_{\pm.12}$ | $\mathbf{0.55}_{\pm.0}$ / $\mathbf{0.55}_{\pm.0}$ | $0.0_{\pm.0}$ / $0.01_{\pm.0}$ | $0.05_{\pm.03}$ / $0.05_{\pm.01}$ | – |
|  | CBM + Seq | $\underline{0.44}_{\pm.05}$ / $0.2_{\pm.2}$ | $0.17_{\pm.03}$ / $0.09_{\pm.01}$ | $\underline{0.5}_{\pm.0}$ / $\underline{0.56}_{\pm.0}$ | $0.05_{\pm.0}$ / $0.01_{\pm.0}$ | $0.05_{\pm.0}$ / $0.08_{\pm.0}$ | $\mathbf{0.95}_{\pm.05}$ / $\mathbf{0.9}_{\pm.02}$ |
|  | CBM + Joint | $0.41_{\pm.04}$ / $0.02_{\pm.01}$ | $0.2_{\pm.0}$ / $0.1_{\pm.0}$ | $0.5_{\pm.0}$ / $0.56_{\pm.0}$ | $0.05_{\pm.0}$ / $0.01_{\pm.0}$ | $0.08_{\pm.03}$ / $0.07_{\pm.0}$ | $\underline{1.0}_{\pm.0}$ / $\underline{0.95}_{\pm.03}$ |

taining the digit pair $(1, 1)$ in red, and add a corresponding *"red"* concept to the concept annotations, and (ii) downsample 10% of the samples with digits $(2, 2)$. This induces two ground-truth error slices: $(1, 1)$ samples without the *"red"* attribute and the underrepresented $(2, 2)$ group.

### 5.2. Baselines

We compare CB-SLICE against three state-of-the-art SDMs: Domino (Eyuboglu et al., 2022), GEORGE (Sohoni et al., 2020), HiBug2 (Chen et al., 2025), and Spotlight (d'Eon et al., 2022). In addition, we include K-Means (MacQueen, 1967) as a baseline. For a fair evaluation, we select the number of slices for Domino, GEORGE, Spotlight, and K-Means using the Silhouette score (Rousseeuw, 1987), with values ranging from 2 to 20; for HiBug2, the number of slices is determined automatically by its enumeration algorithm. See Appendix C for details.

### 5.3. Metrics

**Precision@10:** Proposed by Eyuboglu et al. (2022), this metric measures how accurately a slice discovery method succeeds in recovering the ground-truth error slices. We adopt the formulation used by Yenamandra et al. (2023): Let $\mathcal{S}^* = \{S_1^*, \ldots, S_{t_s}^*\}$ denote the set of ground-truth error slices and $\mathcal{S} = \{S_1, \ldots, S_{t_g}\}$ the set of discovered error slices. For each discovered slice $S_j$, let $O_j = \{o_{j1}, \ldots, o_{j10}\}$ denote the indices of the top-10 samples ranked by their likelihood of belonging to the slice. The similarity between a ground-truth slice $S_i^*$ and a discovered slice $S_j$ is defined as $P_{10}(S_i^*, S_j) = \frac{1}{10} \sum_{r=1}^{10} \mathbb{I}[\mathbf{x}_{o_{jr}} \in S_i^*]$. Each ground-truth slice is matched to the discovered slice with maximum similarity, and *Precision@10* is computed as $\frac{1}{t_s} \sum_{i=1}^{t_s} \max_{j \in \{1, \ldots, t_g\}} P_{10}(S_i^*, S_j)$.

**Matched Ground-Truth Group Frequency (MGF):** MGF measures the proportion of samples from the ground-truth error slice within its matched discovered slice. We use the

same matching between the discovered and ground-truth slices as in *Precision@10*. For each ground-truth group $S_i^*$, let $\alpha(i) = \arg\max_{j \in \{1, \ldots, t_g\}} P_{10}(S_i^*, S_j)$ be its matched discovered slice. The MGF is then $\frac{1}{t_s} \sum_{i=1}^{t_s} \frac{|S_{\alpha(i)} \cap S_i^*|}{|S_{\alpha(i)}|}$.

## 6. Results

### 6.1. Recovery of known bias groups (Q1)

We first examine whether CB-SLICE successfully recovers known failure modes against state-of-the-art baselines. Since CB-SLICE relies on the CBM architecture, it applies only to CBM-based models. For a fair comparison, we evaluate all methods on two CBM variants with different training strategies (sequential vs. joint), as well as on a vanilla DNN (see Appendix D for details).

**CB-SLICE accurately recovers known ground-truth error slices, as measured by *Precision@10* (Table 1, blue).** Across all datasets and model variants, CB-SLICE consistently outperforms competing methods. The only exception is in MetaShift, where it performs on par with GEORGE under sequentially trained CBM. However, for the remaining benchmarks, CB-SLICE surpasses GEORGE by a significant margin (e.g., up to $\sim 65\%$ points on Waterbirds). The high *Precision@10* values also indicate that CB-SLICE identifies slices whose representative samples predominantly belong to the corresponding ground-truth error slices. This advantage likely arises because CB-SLICE forms slices exclusively from erroneous samples, allowing it to focus on isolating failure modes without being confounded by correctly classified ones. Notably, both Spotlight and K-Means perform poorly. One likely reason is that K-Means clusters samples solely based on embedding similarity, without incorporating error-aware signals, while Spotlight relies on high-loss values, which provide insufficient discrimination between different failure modes. We further verify that these results are robust to the choice of backbone encoder in Appendix A.4.

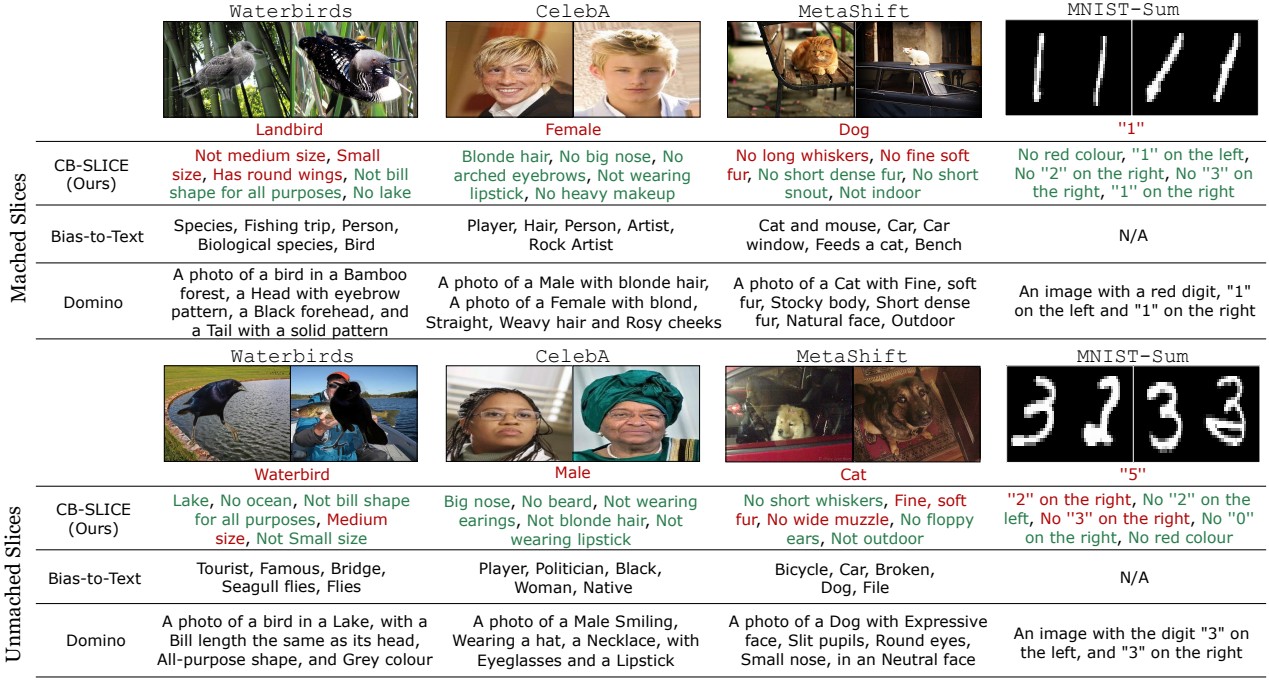

Figure 2. Matched (top) and unmatched (bottom) error slice examples, selected by the SI score (Eq. 10) for each benchmark. For each slice, we show two samples with the highest $P(S_j \mid \mathbf{x})$ and the mispredicted class. CB-SLICE keywords are compared to Bias-to-Text and Domino. Mispredicted concept-keywords are shown in red and correctly predicted ones in green. CB-SLICE captures spurious attributes in matched slices and reveals previously unknown failure modes.

## 6.2. Failure mode alignment (Q2)

Effective SDMs should isolate samples exhibiting the same underlying failure modes, thereby facilitating diagnosis. Hence, we next examine whether the discovered error slices align with coherent, well-defined failure modes. We evaluate slice alignment by measuring the homogeneity of each discovered slice to its corresponding ground-truth slice.

**CB-SLICE discovers homogeneous error slices in terms of MGF (Table 1, pink).** Table 1 shows that CB-SLICE forms error slices that mostly consist of samples from the same corresponding ground-truth error slice, thereby more faithfully capturing the underlying failure mode. We note that while MGF may favour forming multiple smaller, compact slices over fewer larger ones, this emphasis on homogeneity is desirable. This is because it isolates compact, well-characterised sub-failure modes that may be hidden within the broad ground-truth error slice, thereby supporting targeted mitigation strategies. Alternative metrics, such as completeness (Rosenberg & Hirschberg, 2007), on the other hand, quantify how dispersed the samples are, achieving their optimal value when all samples from the same ground-truth slice are grouped together. This can obscure distinct failure modes within the broad ground-truth population (e.g.,

distinct underrepresented subgroups among *"blonde hair"* males, such as *"young"* vs. *"rosy cheeks"* individuals). For this reason, we prioritise slice homogeneity as a more informative and actionable evaluation criterion.

## 6.3. Error explanation through concept predictions (Q3)

We evaluate how faithfully the keyword-concepts produced by CB-SLICE explain the underlying source of model errors. Keyword-concepts convey not only the semantic meaning of the concept but also whether it is predicted to be present and whether this prediction is correct. As a result, a single keyword in CB-SLICE can provide more meaningful diagnostic information than a simple text explanation. We empirically evaluate this hypothesis by analysing slices in two settings: (1) *matched* slices aligned with ground-truth error slices, thereby representing well-defined failure modes, and (2) *unmatched* slices, which do not correspond to any predefined group and instead reflect the discovery of previously unknown failure modes. The lack of a clear ground-truth target for the unmatched slices makes quantitative benchmarking against prior methods inherently challenging. As a result, unmatched slices are evaluated only qualitatively.

We compare CB-SLICE against two error slice explanation

baselines: Bias-to-Text (Kim et al., 2024b) and Domino. For Domino, we evaluate its explanations based on the slices discovered by CB-SLICE. For each method, we report the top-5 explanatory keywords based on the method's scoring mechanism. For CB-SLICE, these are the concepts with the highest ECSA scores (Eq. 7). See Appendix C.2 for details.

**Concept predictions provide meaningful explanations for the underlying failure modes.** Figure 2 demonstrates how concept-based explanations that are directly grounded in the model's prediction process, enable deeper insight into model failures. For instance, in both `Waterbirds` examples, CB-SLICE attributes failures primarily to systematic mispredictions of size-related concepts. Specifically, the bottom example reveals an error slice in which *Landbirds* are misclassified as *Waterbirds*, driven by incorrect prediction of the *"medium size"* concept. Dataset analysis shows that size is a highly discriminative attribute: among samples annotated as *"medium size"*, $\sim 94\%$ belong to the *Waterbirds* class, whereas only about $5\%$ belong to *Landbirds*. When *"medium size"* co-occurs with a *"lake"* background, the likelihood of association with the *Landbirds* class drops to $0\%$ compared to $\sim 100\%$ for *Waterbirds*. This joint configuration of concepts, therefore, strongly biases the model toward the *Waterbirds* class, explaining the observed failure mode. Notably, most *Landbirds-on-water* samples are classified correctly ($\sim 80\%$, Appendix D), indicating that background alone does not determine failure. By exposing the concept predictions that collectively drive misclassification, CB-SLICE provides more precise and faithful explanations of the underlying failure modes.

In the `MNIST-Sum` example in Figure 2 (bottom), CB-SLICE uncovers a previously unknown failure mode: the right-hand digit "3" is systematically misread as "2", yielding a sum of "5" instead of "6". The extracted keywords show that the model both predicts "2" and rules out "3", indicating systematic confusion rather than uncertainty. This suggests that specific visual variants of "3" overlap with "2" in the learnt representation, motivating targeted data augmentation to better separate the classes.

**CB-SLICE provides a clear signal for identifying underrepresented data groups.** CB-SLICE sometimes discovers slices where all keyword-concepts are correctly predicted, indicating that while the concept encoder $g$ accurately captures image attributes, the label predictor $f$ still misclassifies. This typically occurs when a combination of concepts is rare in training, leaving $f$ undertrained on that pattern. For example, in the `MNIST-Sum` case shown in Figure 2 (top), CB-SLICE identifies the underrepresented $(1, 1)$ group without the spurious *"red"* attribute, which constitutes only 10% of all $(1, 1)$ samples. Although each individual concept (e.g., *"1 on the left"*) appears frequently across other groups (e.g., $(1, 2)$), their joint occurrence is

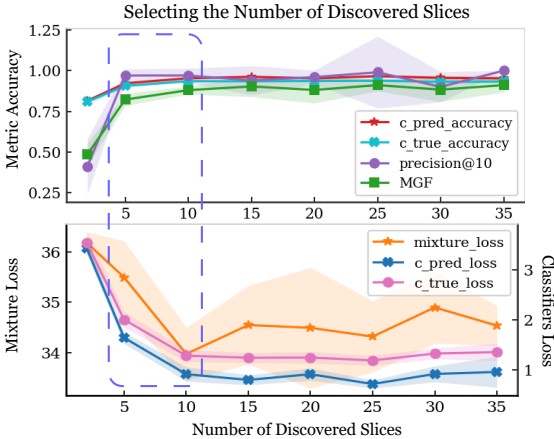

*Figure 3.* CB-SLICE loss components (bottom) and auxiliary classifier accuracies (top) vs. the number of slices $t_g$. *Precision@10* and MGF are overlaid on the accuracy plot for comparison. As highlighted by the blue box, loss convergence aligns with metrics stabilisation, providing a principled criterion for selecting $t_g$.

rare, causing $f$ to mispredict the class "2". A similar pattern appears in the `CelebA` (top) example, where *Male* samples with *"blonde hair"* and *"no big nose"* occur at only $\sim 1.5\%$, compared to $\sim 22\%$ for the corresponding *Female* group, directly explaining the systematic error. These slices are straightforward to diagnose, can characterise rare concept combinations, and guide targeted data augmentation.

Finally, it is important to note that, unlike CB-SLICE, competing methods rely on auxiliary models to generate explanations, which may themselves encode societal biases, thereby compromising explanation fidelity. As illustrated by the `CelebA` female example in Figure 2 (bottom), Bias-to-Text attributes errors to being *"native"*, while Domino attributes them to being *"man"*. Such incorrect explanations can mislead mitigation efforts, thereby perpetuating existing inequalities. These cases highlight the importance of using a model-grounded approach for failure explanation. For more qualitative examples, see Appendix E.

### 6.4. Selecting the number of discovered slices (Q4)

**CB-SLICE offers a principled criterion for selecting $t_g$.** The choice of the number of discovered slices $t_g$ strongly affects slice quality, yet it is often chosen heuristically. While one would ideally select $t_g$ to maximise *Precision@10* and MGF, these metrics require ground-truth slice labels, which are unavailable in practice. We therefore examine whether CB-SLICE provides a principled signal for choosing $t_g$ by analysing the convergence of its loss (Eq. 6) against downstream performance on `MNIST-Sum`. As shown in Figure 3, CB-SLICE's loss converges at the same $t_g$ range ($\sim [5, 10]$) where the accuracies of auxiliary classifiers and evaluation

metrics saturate. This alignment suggests that loss convergence provides a practical criterion for selecting $t_g$.

## 7. Discussion and Conclusion

**Limitations** CB-SLICE relies on CBMs and thus inherits their assumptions, notably the need for complete and faithful concept annotations characterising the data, which can also describe failure modes. Hence, its effectiveness may degrade when concepts are noisy (Park et al., 2026) or incomplete (Yeh et al., 2020; Espinosa Zarlenga et al., 2022). CB-SLICE also requires training a CBM, incurring additional computational cost. Nonetheless, using dedicated models for dataset and failure analysis is increasingly common (Wu et al., 2023; Nam et al., 2020; Yenamandra et al., 2023). Moreover, CBMs are a maturing model family that is narrowing the performance gap with standard DNNs (e.g., Espinosa Zarlenga et al. 2025a), and can be built on top of existing black-box DNNs (e.g., Yuksekgonul et al. 2022). Future work could therefore (1) extend CB-SLICE to operate in settings with incomplete (Shang et al., 2024; Liu et al., 2025), noisy (Penaloza et al., 2025), or unavailable concept sets (Yang et al., 2023; Oikarinen et al., 2023), and (2) integrate it with downstream mitigation to close the loop between failure discovery and model improvement.

**Conclusion** In this work, we show how CBMs provide a principled foundation for error slice discovery. We introduce CB-SLICE, a concept-based SDM that identifies and explains failure modes using CBMs. CB-SLICE reframes error slice discovery as a model-aware process: rather than relying on feature correlations or post-hoc descriptions disconnected from the model's internal logic, it grounds explanations in the representations the model actually learns. We show that CB-SLICE recovers known failure modes and discovers previously unknown ones while providing rich, faithful explanations of error sources. Crucially, CB-SLICE distinguishes distinct failure types, including slices driven by systematic concept mispredictions and slices arising from rare, correctly predicted concept combinations. We hope this work motivates further model-aware approaches to failure discovery and explanation, advancing more reliable and transparent ML systems.

## Impact Statement

This paper presents work aimed at advancing the field of Machine Learning by improving the identification and explanation of systematic model failures. CB-SLICE supports more faithful and interpretable model debugging by grounding error analysis in concept-based model representations. This can support the development of more robust and reliable models, particularly in settings where identifying underrepresented data groups or spurious correlations is important for safe deployment. While CB-SLICE may help practitioners diagnose and mitigate biases more effectively, its applicability depends on the availability and quality of concept annotations. As with all interpretability methods, explanations should be used with appropriate domain expertise and caution. We do not identify any ethical concerns beyond standard considerations in responsible and trustworthy machine learning.

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

# Appendix

## CB-SLICE: Concept-Based Interpretable Error Slice Discovery

### A. Ablation Study

In this section, we present a series of ablation studies examining the contribution of CB-SLICE's key design choices.

#### A.1. Error-Prone Concept Filtering

We first examine the effect of error-prone concept filtering (Section 3.1) on the quality of error slice discovery by comparing the use of the full concept set with a subset of error-prone concepts $C_{\mathrm{err}}$ on `Waterbirds` under the joint CBM training strategy (Figure 4). The subset $C_{\mathrm{err}}$ is constructed by selecting, for each class, the top-10 concepts with the highest ECTP scores (Eq. 2). Across all evaluation metrics, namely Slice Informativeness (SI; Eq. 10), Precision@10, and MGF (Section 5.3), restricting the slice formation process to these error-prone concepts consistently improves performance compared to using the full concept set. These results indicate that focusing on concepts most strongly linked to prediction errors helps isolate more coherent and homogeneous failure modes, leading to more informative and actionable error slices.

**Concept Filtering Ablation Analysis**

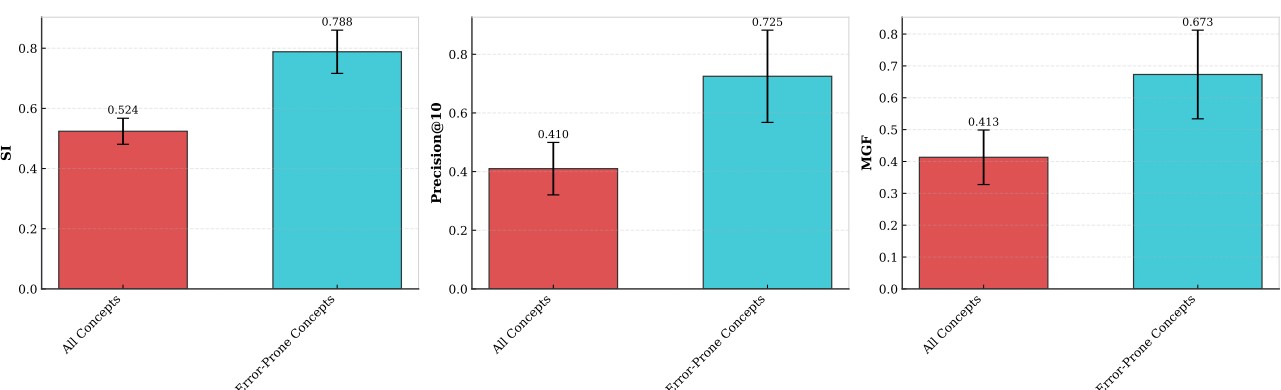

*Figure 4.* Error-prone concept filtering ablation on `Waterbirds` under joint CBM training. We evaluate the impact of restricting slice discovery to the error-prone concept subset $C_{\mathrm{err}}$ compared to using the full concept set. Performance is measured using Slice Informativeness (SI; left), Precision@10 (middle), and MGF (right). Bars report mean values over five runs, with error bars indicating standard deviation. Across all metrics, using the error-prone concept subset yields substantially higher scores than using all concepts, indicating that filtering to concepts most associated with model errors improves the coherence, precision, and homogeneity of the discovered slices.

#### A.2. Loss Components

Here, we study how each loss component in CB-SLICE contributes to the discovered error slice quality via an ablation analysis on `MNIST-Sum` under the joint CBM training strategy (Figure 5). We compare the base GMM likelihood objective (Eq. 3) to variants that additionally incorporate supervision from ground-truth concepts $\mathcal{L}_{C_{\mathrm{true}}}$ (Eq. 5), predicted concepts $\mathcal{L}_{C_{\mathrm{pred}}}$ (Eq. 5), and their combinations. Across all three metrics, Slice Informativeness (SI, Eq. 10), Precision@10, and MGF, adding concept-level supervision consistently improves slice coherence and homogeneity over the GMM-only baseline. The best performance is achieved when $\mathcal{L}_{C_{\mathrm{true}}}$ and $\mathcal{L}_{C_{\mathrm{pred}}}$ are combined with the GMM objective, indicating that jointly modelling semantic concept structure and prediction errors is crucial for discovering coherent, well-characterised failure modes.

#### A.3. GMM vs Linear Classifier

We study the benefits of using a Gaussian Mixture Model (GMM) instead of a simpler linear layer to form error slices. CB-SLICE aims to produce slices that are both semantically coherent and aligned with a shared concept-level error pattern

**Loss Components Ablation Analysis**

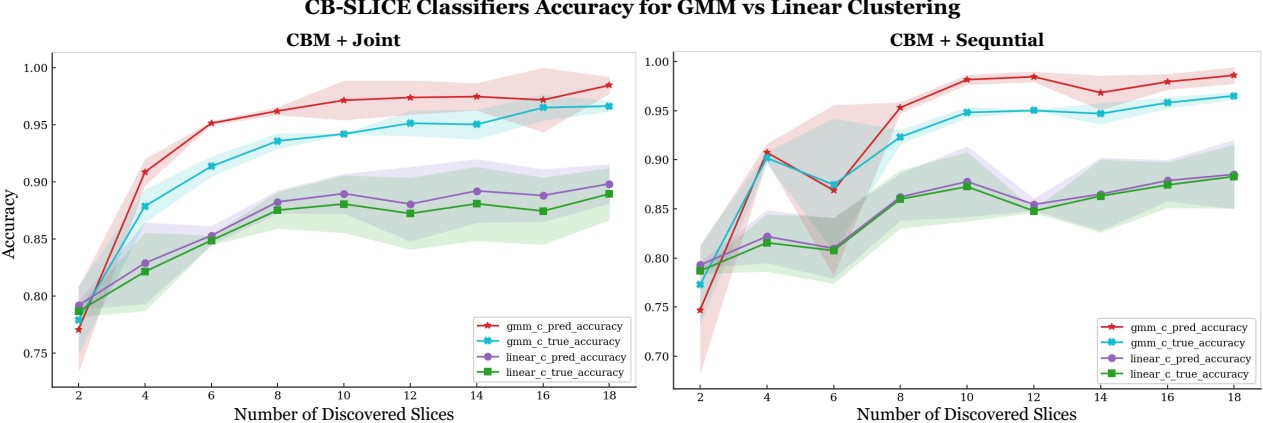

*Figure 5.* Loss components ablation on `MNIST-Sum` under joint CBM training. We evaluate the contribution of each component in the CB-SLICE slicing objective by ablating the GMM likelihood term ($\mathcal{L}_{\text{GMM}}$), and the auxiliary classification terms on ground-truth concepts ($\mathcal{L}_{c_{\text{true}}}$) and predicted concepts ($\mathcal{L}_{c_{\text{pred}}}$), as well as their combinations. Performance is reported in terms of Slice Informativeness score (SI; left), Precision@10 (middle), and slice homogeneity measured by the MGF score (right). Bars show mean values over five runs, with error bars indicating standard deviation. Incorporating both concept-level supervision terms alongside the GMM objective yields the strongest and most consistent improvements across all metrics, highlighting the complementary role of semantic grounding and concept-level supervision in discovering coherent error slices.

**CB-SLICE Classifiers Accuracy for GMM vs Linear Clustering**

*Figure 6.* CB-SLICE auxiliary classifier accuracies, $z_c(\cdot)$ and $z_{\hat{c}}(\cdot)$, for GMM vs. linear clustering. **Left:** Results for the jointly trained CBM. **Right:** Results for the sequentially trained CBM. Each plot shows the accuracy of $z_c(\cdot)$ (*c_true_accuracy*) and $z_{\hat{c}}(\cdot)$ (*c_pred_accuracy*) as a function of the number of discovered slices, comparing GMM-based slicing (red and cyan) to a linear alternative (purple and green) on `MNIST-Sum`. Curves report mean performance over three seeds, with shaded standard deviation. Across both training strategies, GMM-based slicing consistently outperforms linear slicing for both classification tasks, yielding gains of up to about 10% points in accuracy.

in the concepts' logit space. To this end, it incorporates two auxiliary classifiers, $z_c(\cdot)$ and $z_{\hat{c}}(\cdot)$, into the slicing objective. These classifiers predict the ground-truth concept labels and the model concept predictions, respectively, based on the likelihood probability $P(S_j \mid \mathbf{x})$ of sample $\mathbf{x}$ belonging to slice $S_j$.

While GMMs benefit from comprehensive statistical modelling, they can be computationally costly, requiring $2k_{\text{err}} \times t_g + t_g$ additional parameters. As a simpler alternative, one can instead use a linear layer over the embedding space to form slices in the same way by producing $P(S_j \mid \mathbf{x})$, given $H_{\text{err}}^{(\Psi_{\text{val}})}$. Such modelling requires only $k_{\text{err}} \times t_g$ additional parameters.

In Figure 6, we compare these slicing alternatives across two CBM variants (joint and sequential training), evaluating the accuracies of $z_c(\cdot)$ and $z_{\hat{c}}(\cdot)$ over varying numbers of discovered slices $t_g$ and three random seeds on `MNIST-Sum`. Across all settings, GMM-based slicing consistently outperforms the linear alternative, improving the accuracy of each auxiliary classifier by up to $\sim 10$ points. These findings suggest that using a GMM to model the concepts' logit space not only offers

a more expressive statistical representation, but also produces more informative, concept-level error-aligned slices compared to a linear partitioning.

## A.4. Sensitivity to Encoder Choice

Lastly, we evaluate the sensitivity of CB-SLICE to the choice of backbone encoder by replacing the default ResNet-18 with a DenseNet-121 (Huang et al., 2017) on the `Waterbirds` dataset. Table 2 reports Precision@10 and MGF for both encoders across all model variants and baselines. CB-SLICE maintains its advantage over all baselines under both encoders, achieving comparable Precision@10 (e.g., $0.79$ vs. $0.78$ for CBM + Seq and $0.82$ for CBM + Joint) and MGF scores. These results suggest that CB-SLICE's performance is not specific to the ResNet-18 encoder choice and can be generalised across various backbone architectures.

*Table 2.* Sensitivity to encoder choice for `Waterbirds` task. Precision@10 and MGF, reported as mean $\pm$ std over five seeds. Best results per task, and those not statistically significantly different, are **bolded** and underlined.

| ENCODER | MODEL | DOMINO | GEORGE | HiBug2 | SPOTLIGHT | K-MEANS | CB-SLICE (OURS) |
|---|---|---|---|---|---|---|---|
| ResNet-18 | Vanilla DNN | **0.53**$_{\pm.09}$ / 0.05$_{\pm.0}$ | 0.34$_{\pm.04}$ / 0.15$_{\pm.01}$ | 0.25$_{\pm.0}$ / **0.25**$_{\pm.0}$ | 0.05$_{\pm.0}$ / 0.05$_{\pm.0}$ | 0.05$_{\pm.0}$ / 0.05$_{\pm.0}$ | – |
| | CBM + Seq | 0.72$_{\pm.03}$ / 0.17$_{\pm.01}$ | 0.22$_{\pm.1}$ / 0.13$_{\pm.07}$ | 0.2$_{\pm.0}$ / 0.25$_{\pm.0}$ | 0.05$_{\pm.0}$ / 0.04$_{\pm.0}$ | 0.11$_{\pm.04}$ / 0.03$_{\pm.0}$ | **0.78**$_{\pm.07}$ / **0.7**$_{\pm0.05}$ |
| | CBM + Joint | 0.62$_{\pm.04}$ / 0.24$_{\pm.17}$ | 0.18$_{\pm.13}$ / 0.08$_{\pm.03}$ | 0.25$_{\pm.0}$ / 0.25$_{\pm.0}$ | 0.0$_{\pm.0}$ / 0.04$_{\pm.0}$ | 0.09$_{\pm.04}$ / 0.03$_{\pm.0}$ | **0.82**$_{\pm.03}$ / **0.76**$_{\pm.04}$ |
| DenseNet-121 | Vanilla DNN | **0.42**$_{\pm.06}$ / 0.05$_{\pm.0}$ | 0.25$_{\pm.13}$ / 0.08$_{\pm.03}$ | 0.25$_{\pm.0}$ / **0.23**$_{\pm.0}$ | 0.1$_{\pm.0}$ / 0.03$_{\pm.0}$ | 0.0$_{\pm.0}$ / 0.05$_{\pm.0}$ | – |
| | CBM + Seq | 0.51$_{\pm.09}$ / 0.05$_{\pm.0}$ | 0.23$_{\pm.06}$ / 0.15$_{\pm.01}$ | 0.39$_{\pm.0}$ / 0.39$_{\pm.0}$ | 0.0$_{\pm.0}$ / 0.03$_{\pm.0}$ | 0.0$_{\pm.0}$ / 0.03$_{\pm.0}$ | **0.79**$_{\pm.04}$ / **0.61**$_{\pm0.04}$ |
| | CBM + Joint | 0.56$_{\pm.02}$ / 0.16$_{\pm.0}$ | 0.14$_{\pm.13}$ / 0.13$_{\pm.03}$ | 0.25$_{\pm.0}$ / 0.24$_{\pm.0}$ | 0.0$_{\pm.0}$ / 0.03$_{\pm.0}$ | 0.05$_{\pm.0}$ / 0.03$_{\pm.0}$ | **0.82**$_{\pm.08}$ / **0.66**$_{\pm.06}$ |

## B. Datasets

In this section, we describe each dataset used in our experiments and detail the corresponding preprocessing.

**Waterbirds** (Sagawa et al., 2019) is a binary bird classification dataset (*Landbirds* vs. *Waterbirds*) composed of RGB images created by overlaying bird crops from the Caltech-UCSD Birds-200-2011 (CUB) dataset (Wah et al., 2011) onto background images from the Places dataset (Zhou et al., 2017). Backgrounds are drawn from four scene categories: *bamboo forest*, *forest*, *lake*, and *ocean*. This construction yields four subgroups: *Landbirds-on-land*, *Landbirds-on-water*, *Waterbirds-on-land*, and *Waterbirds-on-water*. In the training set, *Landbirds-on-land* and *Waterbirds-on-water* are overrepresented relative to other groups, inducing a spurious correlation between the class label and the background.

We follow the preprocessing of Koh et al. (2020): images are randomly cropped to $3 \times 229 \times 229$, resized to $3 \times 224 \times 224$, and normalised. While the CUB dataset contains 312 binary attribute annotations, we use a subset of 112 binary concepts, following the attribute selection used by Koh et al. (2020), to remain consistent with prior CBM literature. To this concept set, we further add four binary background concepts that correspond to the background categories: *"bamboo forest"*, *"forest"*, *"lake"*, and *"ocean"*.

In the official dataset, the validation and test sets are balanced across subgroups, removing the spurious correlation. Since our goal is to discover failure modes under the training distribution, we merge the original validation and test sets and resample the validation set to match the training distribution. We report the data split used in our experiments in Table 3.

**CelebA** (Liu et al., 2015) is a face attribute dataset consisting of RGB images annotated with 40 binary attributes. We use the *Male* attribute as the task label (*Male* vs. *no Male* as *Female*) and treat the remaining 39 attributes as concepts. Images are resized to $3 \times 224 \times 224$ and normalised. We report the data split used in our experiments in Table 4.

**MetaShift** (Liang & Zou, 2022) We use the *Cat vs. Dog* setup, which aims to address spurious correlations between the *Cat* and *Dog* classes, associated with the *"indoor"* and *"outdoor"* attributes, respectively. To generate the dataset, we follow the official implementation released by the authors [4] and resample the validation set to match the training set distribution. We report the data split used in our experiments in Table 5. For image preprocessing, we follow the same procedure as for `CelebA`. For concept annotations, we adopt the Label-free pipeline of Oikarinen et al. (2023), using the concept bank from Enouen et al. (2025) and a zero-shot CLIP classifier (Radford et al., 2021) to assign binary concept labels for each image.

**MNIST-Sum** is an MNIST-based dataset (LeCun et al., 1998) composed of normalised $1 \times 28 \times 56$ greyscale images

---
[4] https://github.com/Weixin-Liang/MetaShift/tree/main

formed by concatenating two MNIST digits ranging from 0 to 3, inclusive. The label is a categorical number representing the sum of the digits, yielding 7 classes (e.g.,$(1, 2) \mapsto 3$). We define the concept annotations as a binary vector $c \in \{0, 1\}^8$ that encodes the digits: the first four dimensions represent the left-side digit, and the last four represent the right-side digit. For example, the digit pair $(0, 1)$ corresponds to $c = [1, 0, 0, 0, 0, 1, 0, 0]$ with task label 1.

To simulate realistic biases within the dataset, we follow Domino (Eyuboglu et al., 2022) and introduce two types of ground-truth error slice groups: (a) *rare group* where a selected population (e.g., $(2, 2)$) is downsampled to appear with only $10\%$ of the frequency relative to other groups, and (b) *spuriously correlated group* where $90\%$ of samples from a selected group (e.g., (1,1)) are correlated with a spurious attribute (e.g., *"red"*). In addition to the digit concepts, we introduce a ninth binary concept indicating the presence of the *"red"* attribute. We report the data split used in our experiments in Table 6.

*Table 3.* Waterbirds dataset splits.

| DATA SPLIT | LANDBIRD | | WATERBIRD | | TOTAL |
| --- | --- | --- | --- | --- | --- |
| | LAND | WATER | LAND | WATER | |
| # TRAINING SAMPLES | 3,498 | 184 | 56 | 1,057 | 4,795 |
| # VALIDATION SAMPLES | 2,722 | 155 | 44 | 775 | 3,696 |
| # TOTAL | 6,559 | | 1,932 | | 8,491 |

*Table 4.* CelebA dataset splits.

| DATA SPLIT | FEMALE | | MALE | | TOTAL |
| --- | --- | --- | --- | --- | --- |
| | BLONDE | NON-BLONDE | BLONDE | NON-BLONDE | |
| # TRAINING SAMPLES | 22,880 | 71,629 | 1,387 | 66,874 | 162,770 |
| # VALIDATION SAMPLES | 2,874 | 8,535 | 182 | 8,276 | 19,867 |
| # TOTAL | 118,165 | | 84,434 | | 202,599 |

*Table 5.* MetaShift dataset splits.

| DATA SPLIT | CAT | | DOG | | TOTAL |
| --- | --- | --- | --- | --- | --- |
| | INDOOR | OUTDOOR | INDOOR | OUTDOOR | |
| # TRAINING SAMPLES | 745 | 150 | 150 | 745 | 1,790 |
| # VALIDATION SAMPLES | 249 | 50 | 50 | 249 | 598 |
| # TOTAL | 1,194 | | 1,194 | | 2,388 |

*Table 6.* MNIST-Sum dataset splits.

| DATA SPLIT | TOTAL 0 | 1 | | 2 | | | 3 | | | 4 | | 5 | | 6 | TOTAL |
| --- | --- | --- | --- | --- | --- | --- | --- | --- | --- | --- | --- | --- | --- | --- | --- | --- |
| | (0,0) | (0,1) | (1,0) | (0,2) | (1,1) | (1,1, RED) | (2,0) | (0,3) | (1,2) | (2,1) | (3,0) | (1,3) | (2,2) | (3,1) | (2,3) | (3,2) | (3,3) | |

| DATA SPLIT | (0,0) | (0,1) | (1,0) | (0,2) | (1,1) | (1,1, RED) | (2,0) | (0,3) | (1,2) | (2,1) | (3,0) | (1,3) | (2,2) | (3,1) | (2,3) | (3,2) | (3,3) | TOTAL |
| --- | --- | --- | --- | --- | --- | --- | --- | --- | --- | --- | --- | --- | --- | --- | --- | --- | --- | --- |
| # TRAINING SAMPLES | 250 | 250 | 250 | 250 | 25 | 225 | 250 | 250 | 250 | 250 | 250 | 250 | 25 | 250 | 250 | 250 | 250 | 3775 |
| # VALIDATION SAMPLES | 250 | 250 | 250 | 250 | 25 | 225 | 250 | 250 | 250 | 250 | 250 | 250 | 25 | 250 | 250 | 250 | 250 | 3775 |
| # TOTAL | 500 | 1000 | | 1500 | | | 2000 | | | | | 1050 | | | 1000 | | 500 | 7550 |

# C. Slice Discovery Methods: Implementation Details

## C.1. CB-SLICE

**Error-prone concept set.** As described in Section 3.1, CB-SLICE forms error slices using a subset of *error-prone concepts* $C_{\text{err}}$. In our experiments, we construct this subset by selecting, for each class, the top $t_e = 10$ concepts with the highest ECTP scores (Eq. 2). This choice yields a focused set of concepts that contribute most strongly to downstream errors. While $t_e$ can be adjusted depending on the task (e.g., increased when more concept annotations are available), we adopt a moderate value to balance explanatory power with computational efficiency.

**Training details:** For each benchmark, we tune the number of discovered slices $t_g$ and the trade-off hyperparameter $\lambda$, and select the values that maximise the auxiliary classifiers' accuracies, as discussed in Section 6.4. We report the selected

*Table 7.* Selected hyperparameters for CBM trained jointly (**Joint**) and sequentially (**Seq**) across datasets.

| Hyperparameters | Waterbirds | | CelebA | | MetaShift | | MNIST-Sum | |
|---|---|---|---|---|---|---|---|---|
| | Joint | Seq | Joint | Seq | Joint | Seq | Joint | Seq |
| $t_g$ | 10 | 10 | 8 | 16 | 10 | 7 | 15 | 10 |
| $\lambda$ | 5 | 10 | 5 | 40 | 20 | 20 | 15 | 15 |

*Table 8.* Number of discovered slices selected for each baseline method by maximising the Silhouette score. Baselines: Domino (D), GEORGE (G), Spotlight (S), and K-Means (K).

| Model variant | Waterbirds | | | | CelebA | | | | MetaShift | | | | MNIST-Sum | | | |
|---|---|---|---|---|---|---|---|---|---|---|---|---|---|---|---|---|
| | **D** | **G** | **S** | **K** | **D** | **G** | **S** | **K** | **D** | **G** | **S** | **K** | **D** | **G** | **S** | **K** |
| **Vanilla DNN** | 4 | 4 | 3 | 4 | 7 | 4 | 2 | 4 | 14 | 4 | 3 | 4 | 8 | 6 | 2 | 9 |
| **CBM + Joint** | 4 | 4 | 2 | 4 | 7 | 4 | 3 | 4 | 5 | 4 | 2 | 4 | 4 | 4 | 2 | 15 |
| **CBM + Seq** | 4 | 6 | 3 | 4 | 2 | 4 | 2 | 2 | 6 | 4 | 3 | 4 | 5 | 4 | 2 | 14 |

values for each benchmark in Table 7. We train the GMM module for 200 epochs using SGD with an initial learning rate of $0.1$, decayed by a factor of 2 every 30 epochs, and a batch size of 8.

### C.2. Baselines

**Quantitative Baselines.** We compare CB-SLICE against four state-of-the-art SDMs: Domino (Eyuboglu et al., 2022), GEORGE (Sohoni et al., 2020), HiBug2 (Chen et al., 2025), and Spotlight (d'Eon et al., 2022). For Domino and Spotlight, we use the publicly available implementation released by Domino's authors [5]. For HiBug2, we use the authors' official implementation[6], with two adaptations for fair evaluation: (1) we use ground-truth concept labels for its slice enumeration algorithm rather than a VLM-based algorithm, and (2) we enumerate slices jointly across all samples rather than per-class, as per-class enumeration yields slices too small for Precision@10 to be meaningful. Concretely, HiBug2's enumeration algorithm systematically searches over all concept value combinations of up to three concepts, retaining those where model accuracy falls below the dataset average.

**Number of discovered slices.** For Domino, GEORGE, Spotlight, and K-Means, we select the number of discovered slices by maximising the Silhouette score (Rousseeuw, 1987) over the range $[2, 20]$ as reported in Table 8. For HiBug2, the number of slices is determined automatically by its enumeration algorithm.

**Qualitative Baselines.** For the qualitative assessment of slice explanations, we compare CB-SLICE with Bias-to-Text (Kim et al., 2024b) and Domino.

**Bias-to-Text** first produces image-level descriptions via ClipCap (Mokady et al., 2021), then applies the YAKE keyword extraction algorithm (Campos et al., 2020) separately to correctly predicted samples $\mathcal{D}^l_{\text{correct}}$ and misclassified samples $\mathcal{D}^l_{\text{wrong}}$ from each class $l$. To detect class-specific biased keywords, it computes a CLIP-based score that quantifies the similarity between candidate keywords and the misclassified set $\mathcal{D}^l_{\text{wrong}}$, while enforcing dissimilarity to the correctly classified set $\mathcal{D}^l_{\text{correct}}$. In our experiments, we adapt this scoring procedure for each error slice $S_j$ so that $\mathcal{D}^l_{\text{wrong}}$ corresponds to the subset of misclassified samples from class $l$ that fall within slice $S_j$, whereas $\mathcal{D}^l_{\text{correct}}$ contains all correctly classified samples from class $l$. Finally, we choose the top-5 highest-scoring keywords across all classes within each slice as the representative keywords for that slice. For keyword generation, We use the authors' official implementation[7].

**Domino** explains error slices by identifying the description whose CLIP embedding is best aligned with the slice prototype, given a corpus of textual descriptions that characterise the dataset. Concretely, for each slice $S_j$ and class $l$, it computes their prototype embeddings, $\bar{z}_j$ and $\bar{z}^{(l)}$, respectively. Then, for each embedded text description $z^i_{\text{text}}$, it evaluates its relevance to slice $S_j$ via the similarity score $z^i_{\text{text}}{}^T (\bar{z}_j - \bar{z}^{(l)})$, where $l$ is the most frequent class within $S_j$. The sentence with the highest

---

[5] https://github.com/HazyResearch/domino
[6] https://github.com/cure-lab/HiBug2
[7] https://github.com/alinlab/b2t?tab=readme-ov-file

score is selected as the representative for that slice. This scoring scheme encourages the selected description to capture a pattern specific to the slice rather than merely reflecting the dominant class it contains. To obtain these representative descriptions, we use the authors' official implementation [8].

To construct a text description corpus, we utilise the concept set used by CB-SLICE, and embed each concept within a natural language sentence compatible with CLIP queries. For the `Waterbirds` dataset, we use templates of the form *"A photo of a bird with XXX"* for bird part concepts and *"A photo of a bird in XXX"* for background concepts. For `CelebA`, we generate sex-conditioned descriptions for each concept, e.g., *"A photo of a male with blonde hair"* and *"A photo of a female with blonde hair"*. Similarly, for `MetaShift`, we construct sentences following *"A photo of a cat with XXX"* and *"A photo of a dog with XXX"*; for the *"indoor"* and *"outdoor"* concepts, we omit the word "with". For `MNIST-Sum`, we use templates such as *"An image with the digit XXX on the left/right"* and include *"An image of a red digit"* for the colour attribute.

## D. Concept Bottleneck Model Selection and Training Details

**CBM architecture.**    For the CBM architecture, we use a pre-trained ResNet-18 (He et al., 2016) backbone as the concept encoder $g$, for all tasks except `MNIST-Sum`. For `MNIST-Sum`, we adopt a lightweight backbone consisting of two convolutional layers with 32 and 64 channels, respectively, each followed by a ReLU activation, and a fully connected (FC) layer with ReLU. For the label predictor $f$, we use a single FC layer across all tasks. In practice, $f$ can receive as input either *soft* concept probabilities (i.e., $\{p(c_i = 1 \mid \mathbf{x})\}_{i=1}^{k}$) or *hard* binary concept predictions (i.e., $\{\mathbb{I}\left[p(c_i = 1 \mid \mathbf{x}) \geq 0.5\right]\}_{i=1}^{k}$). Prior work shows that hard concept representations reduce *information leakage* (Mahinpei et al., 2021), where additional unintended information is exploited by $f$. Since our goal is to explain the model's failure modes using concept predictions, faithful alignment between concepts and their semantic meanings is essential. We therefore train $f$ using hard concept representations.

**Vanilla DNN architecture.**    For the vanilla DNN baseline, we use the same backbone architectures as above, without the additional label predictor.

**Training Procedure.**    We train all models using stochastic gradient descent with a learning rate of 0.01, a weight decay of $1 \times 10^{-5}$, and a batch size of 32, unless otherwise stated.

For each dataset, we train three model variants: (1) CBM + Seq, trained sequentially; (2) CBM + Joint, trained end-to-end; and (3) Vanilla DNN.

For `Waterbirds`, `CelebA`, and `MetaShift`, we train CBM + Seq by first optimising the concept encoder $g$ for 300 epochs, followed by the label predictor $f$ for an additional 150 epochs, with the learning rate halved every 40 epochs. We train CBM + Joint end-to-end for 400 epochs, halving the learning rate every 60 epochs. The vanilla DNN is trained for 25 epochs, halving the learning rate every 5 epochs.

For `MNIST-Sum`, CBM + Seq, we train the concept encoder $g$ for 20 epochs and the label predictor $f$ for 15 epochs, using an initial learning rate of 0.1 halved every 10 epochs. We train CBM + Joint for 50 epochs with the same learning rate schedule. The vanilla DNN is trained for 50 epochs, halving the learning rate every 10 epochs.

**Performance Benchmarks.**    We report the validation performance across subpopulations in Table 10 for `Waterbirds`, Table 9 for `CelebA`, Table 11 for `MetaShift`, and Table 12 for `MNIST-Sum`. Across all datasets, we observe a clear performance gap between the ground-truth error slices and the remaining groups.

**Latency.**    For all experiments, we use the NVIDIA RTX 8000 GPU. Training a CBM typically takes about 3ms for a single batch iteration (size 32) with an input size $3 \times 224 \times 224$.

## E. Additional Qualitative Results

Figure 7 shows further qualitative examples of error slices identified by CB-SLICE. In line with previous findings, CB-SLICE reveals both well-known biases and newly uncovered failure modes that are not explicitly labelled in the dataset,

---

[8] https://github.com/HazyResearch/domino

*Table 9.* Validation accuracy (%) across `CelebA` subpopulations, with ground-truth error slices **bolded**.

| MODEL | FEMALE | | MALE | |
|---|---|---|---|---|
| | WEARING LIPSTICK | NOT WEARING LIPSTICK | BLONDE | NON-BLONDE |
| VANILLA DNN | 99.8 | **95.1** | **95.6** | 98.9 |
| CBM + SEQ | 98.3 | **68.4** | **76.9** | 96.6 |
| CBM + JOINT | 98.2 | **70.7** | **69.8** | 97.0 |

*Table 10.* Validation accuracy (%) across `Waterbirds` subpopulations, with ground-truth error slices **bolded**.

| MODEL | LANDBIRD | | WATERBIRD | |
|---|---|---|---|---|
| | LAND | WATER | LAND | WATER |
| VANILLA DNN | 99.4 | **83.2** | **75.0** | 96.1 |
| CBM + SEQ | 99.9 | **81.9** | **47.7** | 92.1 |
| CBM + JOINT | 99.7 | **80.6** | **63.6** | 94.7 |

*Table 11.* Validation accuracy (%) across `MetaShift` subpopulations, with ground-truth error slices **bolded**.

| MODEL | CAT | | DOG | |
|---|---|---|---|---|
| | INDOOR | OUTDOOR | INDOOR | OUTDOOR |
| VANILLA DNN | 96.4 | **68.0** | **76.0** | 96.0 |
| CBM + SEQ | 93.2 | **56.0** | **48.0** | 94.4 |
| CBM + JOINT | 96.0 | **38.0** | **42.0** | 94.8 |

*Table 12.* Validation accuracy (%) across `MNIST-Sum` subpopulations, with ground-truth error slices **bolded**.

| MODEL | 0 | 1 | | 2 | | | 3 | | | 4 | | 5 | | 6 |
|---|---|---|---|---|---|---|---|---|---|---|---|---|---|---|
| | (0,0) | (0,1) | (1,0) | (0,2) | (1,1) | (1,1, RED) | (2,0) | (0,3) | (1,2) | (2,1) | (3,0) | (1,3) | (2,2) | (3,1) | (2,3) | (3,2) | (3,3) |
| VANILLA DNN | 98.8 | 96.0 | 95.6 | 92.8 | **35.7** | 100.0 | 92.8 | 87.0 | 90.8 | 88.0 | 90.8 | 93.6 | **0.0** | 92.0 | 93.2 | 92.0 | 84.8 |
| CBM + SEQ | 98.8 | 99.2 | 98.4 | 97.2 | **0.0** | 100.0 | 98.4 | 99.2 | 97.6 | 97.6 | 98.0 | 98.4 | **0.0** | 95.6 | 98.8 | 95.2 | 93.2 |
| CBM + JOINT | 98.8 | 99.6 | 98.0 | 97.2 | **0.0** | 100.0 | 98.4 | 99.6 | 98.4 | 98.0 | 98.8 | 98.4 | **0.0** | 96.0 | 98.4 | 96.0 | 95.2 |

while also offering fine-grained, concept-level explanations for these failures.

For instance, in the `Waterbirds` example in Figure 7 (top), CB-SLICE identifies an error slice in which *Landbirds* are misclassified as *Waterbirds*. CB-SLICE reveals that the model's poor performance on the *Landbirds-on-water* slice cannot be explained by the spurious *water* background alone (e.g., the *"no forest"* concept). While baseline methods highlight background-related attributes (e.g., *"beach"*, *"ocean"*), CB-SLICE uncovers a more concrete failure mechanism: a combination of mispredicted concepts, such as *"medium size"*, *"no black bill"*, and *"no small size"*, whose joint occurrence is unlikely under the true *Landbird* class $P(y = Landbird \mid c_{\text{medium}}, c_{\text{no\_black\_bill}}, c_{\text{no\_small}}) \approx 0$, while the likelihood for the *Waterbird* class increases to $\approx 0.15$. By identifying interacting concept predictions that collectively drive misclassification, CB-SLICE provides more precise and faithful explanations of the underlying failure modes.

For `MNIST-Sum` (top), CB-SLICE reveals an underrepresented group of digit pairs $(2, 2)$ for which the model correctly predicts both digit concepts (i.e., *"2" on the left* and *"2" on the right*) but still mispredicts the target label as "5" instead of "4". This indicates a failure in the label predictor $f$ due to insufficient samples in the dataset with $(2, 2)$ digit pairs. In the `MNIST-Sum` example (bottom), CB-SLICE uncovers a previously unknown failure mode of digit pairs $(0, 2)$, where the model predicts the sum as "0" instead of "2". The extracted keyword-concepts reveal a strong and systematic misinterpretation of the right-hand digit, highlighted by the joint signals of *"no 2 on the right"* and *"0 on the right"*. This indicates high confidence in an incorrect concept prediction, providing a clear and actionable explanation for the failure.

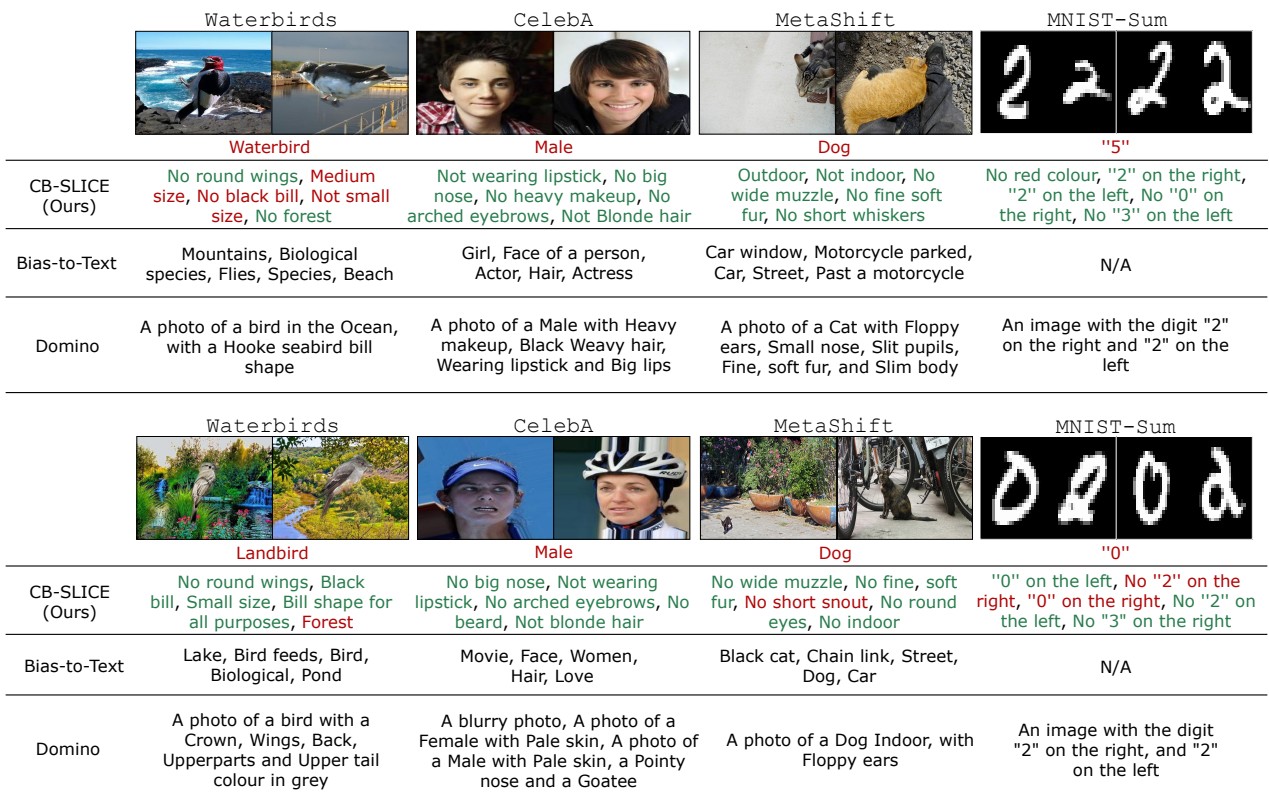

*Figure 7.* Error slice examples, discovered by CB-SLICE. For each slice, we show two samples with the highest $P(S_j \mid \mathbf{x})$ and the mispredicted class. CB-SLICE keywords are compared to Bias-to-Text and Domino. Mispredicted concept-keywords are shown in red and correctly predicted ones in green.

