# OpenReview forum: "CB-SLICE: Concept-Based Interpretable Error Slice Discovery"
_ICML.cc/2026/Conference — ICML 2026 regular_

### Official Review · Reviewer_GYQG · 2026-02-24

**Soundness:** 2
**Presentation:** 2
**Significance:** 3
**Originality:** 3
**Overall Recommendation:** 5
**Confidence:** 3

**Summary:**

The paper introduces CB-SLICE, a error slice discovery method leveraging Concept Bottleneck Models. CB-SLICE groups samples based on having similar failures in the concept prediction and identifies the concepts responsible. The method first identifies error-prone concepts that contribute to downstream task errors. Second, it clusters erroneous samples in the concepts' logit space. Finally, it explains each slice by identifying the core concepts forming it. The paper evaluates the method on 4 datasets, showing high performance compared to the state-of-the-art.

_Note: I am not very familiar with error slice discovery. This makes it hard for me to evaluate the soundness of the experiments and metrics used therein. I've focused on the method itself._

**Compliance With Llm Reviewing Policy:**

Affirmed.

**Final Justification:**

The authors have addressed my concerns. Thus, I recommend acceptance.

**Key Questions For Authors:**

- Question 1: (Equation 1) what is c_hat_i? Is this the concept prediction for concept i? Is it a probability? The symbol should be explained in the text.
- Question 2: The symbol S_{++} seems not explained in the text; I assume it means the set of positive definite matrices (line 178, column 1), but I think many readers would not know this. It would improve readability to explain the symbol.
- Question 3: Can you rephrase "in the concept embedding space" (line 92, column 1), as you are not using concept embeddings [1], instead using the concepts' logit space. The current wording may confuse the reader.
- Question 4: (Equation 5) what is the intuition behind these two losses? What does each of them contribute?
- Question 5: (lines 165-..., column 2 and Equation 7) The explanatory keyword-concepts of a slice are those concepts that induce the largest change in slice membership. But how does this explain a change in task prediction? Changing slice may keep the task prediction the same? So the explanation isn't really explaining the mistake in a task prediction?
- Question 6: Equation 8 seems very arbitrary at first sight, but after doing some derivations I believe p_j^(l) approximates P(y=l | S_j), which would make way more sense than an arbitrary equation. If this is true, it would improve the soundness of the work to really state and explain this.
- Question 7: (Equation 8) y_hat seems unexplained in the text.
- Question 8: (Equation 9) why use cosine similarity to define the compactness of a slice? Wouldn't e.g. dot product make more sense in concept logic space? Again, the reasoning behind the choice of this metric in the text would improve soundness and presentation.

[1] Espinosa Zarlenga et al., Concept Embedding Models.

**Limitations:**

yes

**Strengths And Weaknesses:**

- Strengths
    - Novel and interesting to exploit CBMs for something they're currently not used for, like error slice discovery.
    - Strong results on many datasets. The method systematically outperforms competitors.
- Weaknesses
    - Not all equations are clear as symbols are not always explained (see Questions 1, 2 and 7).
    - It's not easy to understand the reasoning behind some of the components of the model (see Questions 4 and 6), and some choices seem questionable (see Questions 5 and 8). It would be better to explain the reasoning and intuition more in the text.
    - The method seems to only work for soft CBMs (i.e. CBMs where concept predictions aren't binarized), and not for hard CBMs. For instance, Equation 1 would not work with hard CBMs, I believe.

---

> ### Author Rebuttal · Authors · 2026-03-30
>
> Thank you for your very valuable and constructive feedback. We are happy that you found our methodology “novel and interesting” and our results “strong”.
>
> Below, we address your questions. If you have further concerns, please let us know. Otherwise, we would sincerely appreciate it if you would consider updating your score in light of our replies.
>
> ## (Q1, Q2, Q7) Variable definitions
>
> We apologise for any confusion. We follow the notation in previous works (e.g., [1]), but agree that our notation left some variables undefined. To answer your questions:
> 1. **(Q1)** $\hat{c}_i \in [0, 1]$ is the probability predicted for the $i$-th concept by the CBM. We update section 3.1 to explicitly clarify this.
> 2. **(Q2)** Your intuition is correct. We will update Section 3.2 to clarify this.
> 3. **(Q7)** We define $\hat{y}$ as the CBM’s task prediction in line 135, col 2. Nevertheless, we will re-emphasise its definition before Eq. 8 for clarity.
>
> ## (Q3) Use of “Concept Embeddings”
>
> Good point! We will update our manuscript to use “concepts’ logit space” instead.
>
> ## (Q4) Intuition behind losses of Eq. 5
>
> We want to find data slices such that all samples in the same slice have the same ground-truth (gt) concepts and concept predictions, assuming that task mispredictions originate from concept mispredictions in CBMs (e.g., misprediction of $c_\text{blonde}$ causes systematic gender misclassification). The losses in ​​Eq. 5 encourage this by making the slice identity predictive of both the gt and the predicted concepts.
>
> For example, the auxiliary classifier $z_c(\mathbf{r})$ maps the slice probabilities $\mathbf{r} \in [0, 1]^{t_g}$ assigned to a sample $\mathbf{x}$ to its corresponding gt concepts $\mathbf{c}$. The top loss in Eq. 5 thus captures how well slice identity alone can be a predictive signal for $\mathbf{c}$, encouraging members within the same slice to share the same gt concepts. The bottom loss applies the same idea to the concept predictions.
>
> We motivate these loss terms in lines 190-195, col. 1 and will revise it to clarify the point illustrated here. We will also add a new appendix with **[this figure](https://anonymous.4open.science/r/anon-reb/loss_ablation.pdf)** showing the contribution of each loss term to the slice discovery on MNIST-SUM.
>
> ## (Q5) Relationship between change in slice membership and task prediction
>
> When explaining a slice using concepts, we look at the concepts that are “most meaningful” for that slice. When we say we select keyword-concepts based on the concepts whose perturbation induces the largest change in slice membership, we are capturing that notion. In other words, the concepts that are most likely to change our clustering membership prediction from $S_j$ to $S_i$ when we modify their values are the concepts we will prioritise to explain a slice.
>
> Note this is separate from any task errors that samples of a slice may share. We are not trying to explain a “*change* in task prediction” as asked. We are trying to explain, using known keywords, which concepts in a given slice are most responsible for that slice being clustered together by CB-SLICE (which we then use to explain the task error common in that slice).
>
> We will update Section 3.3 to clarify this distinction.
>
>
> ## (Q6) Clarification of Eq. 8
>
> Thank you for this suggestion! Your intuition is correct in that $p_j^{(l)}$ will approximate $P(\hat{y} = l | S_j)$. However, this is a “soft” approximation, as our estimate considers the probability assigned to each slice, even if it's not the slice with the highest probability.
>
> This view of $p_j^{(l)}$ is certainly helpful, and will update our description of MC to provide this intuition.
>
>
> ## (Q8) Choice of Similarity Metric
>
> We chose cosine similarity as it yields a **bounded** similarity metric, whereas dot products do not. Moreover, the dot product is not necessarily the right choice. To see this, assume the centroid’s concept logits are $m \in \mathbb{R}^k$ and, for a sample $\mathbf{x}$ with predicted concept logits $\mathbf{h}$, consider what happens if we scale $\mathbf{h}$ by a factor of $s \in \mathbb{R}^+$ to get $\mathbf{h}^\prime$.
>
> If the concept predictions for $\mathbf{x}$ are saturated around $0$ or $1$ (as is typical for cross-entropy-trained CBMs), then, because the sigmoid function is also saturated at its extremes, it is likely that $\hat{\mathbf{p}} = \text{sigmoid}(\mathbf{h}) \approx  \text{sigmoid}(\mathbf{h}^\prime)$. This means the concept predictions remain approximately the same after scaling. Yet, if we use the dot product to estimate how closely these concept predictions match the centroid predictions, **the output changes** by a factor of $s$. This is undesirable, as we aim to capture coherence across concept predictions.
>
> To clarify this choice, we will update our description of Eq. 9 to explain why we chose a magnitude-agnostic similarity metric.
>
> —
> ## References
> 1. Espinosa Zarlenga et al. "Concept Embedding Models." NeurIPS (2022).

---

> > ### Author Rebuttal · Reviewer_GYQG · 2026-04-01
> >
> > I thank the authors for responding to my remarks. I will increase my score to Accept.

---

### Official Review · Reviewer_tEpV · 2026-03-12

**Soundness:** 3
**Presentation:** 3
**Significance:** 3
**Originality:** 3
**Overall Recommendation:** 4
**Confidence:** 1

**Summary:**

Existing Error Slice Discovery Methods (SDMs) typically rely on external language models or descriptive models to generate explanations for error slices. However, such explanations are often disconnected from the actual reasoning process of the analyzed model. To address this issue, the authors introduce CB-SLICE (Concept-Based Interpretable Error Slice Discovery), a method inspired by Concept Bottleneck Models (CBMs).

**Compliance With Llm Reviewing Policy:**

Affirmed.

**Final Justification:**

My concerns have been adequately addressed. Therefore, I will maintain my positive score.

**Key Questions For Authors:**

Please see Weaknesses.

**Limitations:**

yes

**Strengths And Weaknesses:**

Strengths

1. The paper is generally well written and clearly organized.
2. The proposed method shows a reasonable level of novelty. Introducing Concept Bottleneck Models (CBMs) into the SDM framework is an interesting idea.
3. Experiments are conducted on several benchmarks. The results demonstrate that CB-SLICE achieves higher accuracy in identifying known bias patterns, while also providing more faithful explanations.



Weaknesses

1. Most of the baselines appear to be methods proposed before 2022. It would be helpful to include more recent approaches. In addition, the number of baselines is relatively limited.
2. How can the correctness of Error-Prone Concepts Filtering be ensured? If this step makes mistakes, what impact would it have on the subsequent prediction and slice discovery results?
3. The clarity of the figures could be further improved.
4. How is the number of concepts determined in the experiments?

---

> ### Author Rebuttal · Authors · 2026-03-30
>
> Thank you for your very valuable feedback. We are glad you found our paper “well written and clearly organized” and that it shows “a reasonable level of novelty”.
>
> Below, we address the questions and concerns raised in your review. If you have further concerns, please let us know. Otherwise, we would sincerely appreciate it if you would consider updating your score in light of these replies.
>
> ## (W1) Validity of baselines
>
> Thank you for bringing up this point. First, we would like to point out that our qualitative results are compared against slice explanations generated by Bias-to-Text [1], a baseline introduced in 2024. Bias-to-Text is characteristic of most work in SD over the last couple of years: most recent works have focused on assigning semantics to slices discovered by an underlying method (e.g., Domino), using, for example, foundation models. Instead, we take a different approach and propose a way to avoid costly foundation models by using CBMs.
>
> Second, we would like to point out that although Domino and Spotlight are not recent baselines by AI standards, they remain very strong baselines for evaluating new SD approaches (or related methods). For example, LADDER [2], a recent ACL paper, evaluates its discovery on two baselines, one of which is Domino and the other, FACTS, a paper that came out in 2023 and whose performance, as seen in that evaluation, is not drastically different from Domino (we do not evaluate against LADDER because its use of an LLM in the pipeline would make it an unfair comparison against CB-SLICE).
>
> We nevertheless acknowledge that there is a limited number of more recent baselines we can compare against. Hence, we ran our evaluation on HiBug2 [3]. For evaluation fairness, we adapted HiBug2 so that (1) it uses concept labels for its slice enumeration algorithm, rather than a VLM-based algorithm, and (2) we enumerate slices jointly, as per-class enumeration would yield too-small slices for Precision@10 to be meaningful when minority slices are very small. Our results, shown **[here](https://anonymous.4open.science/r/anon-reb/hibug2.pdf)**, indicate that CB-SLICE outperforms HiBug2 across all evaluation metrics.
>
>
> We will incorporate this into our manuscript by: (1) updating Section 5.2 to discuss why we do not include some recent baselines in our evaluation (as argued above), and (2) including our HiBug2 results in Table 1.
>
>
> ## (W2) Correctness of Error-Prone Concepts Filtering
>
> Regarding this concern, it would be helpful to clarify what is meant by “correctness” in the context of filtering concepts. We nevertheless believe it is a natural question to wonder whether this filtering step is needed in the first place.
>
> At a high level, concept filtering is used for three reasons: (1) it helps by reducing the number of potential concepts we will have to reason about; (2) it makes our clustering approach faster; and (3) it helps our error-aware clustering by removing potential confounding/irrelevant concepts in its feature space.
>
> More concretely, **[this figure](https://anonymous.4open.science/r/anon-reb/waterbirds_filtering.pdf)** shows the results of running CB-SLICE with and without the error-prone concept filtering step on Waterbirds. There, we clearly see that when this filtering is introduced, our method is much better at identifying ground-truth error slices across all metrics (i.e., giving all concepts to the clustering actually hurts the identifiability of correct error slices). These results strongly suggest that there is a benefit to introducing filtering beyond computational gains.
>
> To clarify this for future readers, we will include the new ablation results discussed here as a new appendix in our updated manuscript.
>
> ## (W3) Figure clarity
>
> We are more than happy to update any figures that are unclear. However, if you have a specific figure(s) in mind that you found unclear, it would be helpful to know so that we can address it.
>
> ## (W4) How is the number of concepts determined in the experiments?
>
> The number of concepts is not determined by us; rather, it is a property of the training sets (e.g., CelebA has 39 attribute/concept annotations). We note that this is a common setup assumption in the concept-based modelling literature.
>
> If you meant to ask how the number of slices is determined in our evaluation, we specify our selection process in Section 6.4. To summarise: we select the number of clusters $t_g$ and the trade-off hyperparameter $\lambda$ by performing a grid search over candidate parameters and selecting the ones that maximise the auxiliary classifiers' accuracies on the validation set.
>
> —
>
> ## References
>
> 1. Kim et al. "Discovering and mitigating visual biases through keyword explanation." CVPR (2024).
> 2. Ghosh et al. "Ladder: Language-driven slice discovery and error rectification in vision classifiers." ACL (2025).
> 3. Chen et al. "Hibug2: Efficient and interpretable error slice discovery for comprehensive model debugging." ICLR (2025).

---

> > ### Author Rebuttal · Reviewer_tEpV · 2026-04-02
> >
> > Thank you for the authors’ response. I will maintain my positive score.

---

### Official Review · Reviewer_tXeA · 2026-03-13

**Soundness:** 2
**Presentation:** 3
**Significance:** 2
**Originality:** 2
**Overall Recommendation:** 4
**Confidence:** 3

**Summary:**

This paper presents CB-SLICE, a concept-based slice discovery method designed to identify subpopulations where Concept Bottleneck Models (CBMs) fail systematically. Unlike prior slice discovery approaches that often rely on post-hoc correlations and may not reflect the model's true error source, CB-SLICE exploits the interpretable concept layer of CBMs, where final prediction errors are often caused by mistakes in concept predictions. By grouping samples that share similar concept-level failures, the method discovers error slices with more faithful and fine-grained explanations, and further identifies te key concepts most responsible for each slice's failure mode. Experiments across multiple benchmarks show that CB-SLICE is effective at uncovering known biases and provides richer and more reliable explanations of model errors than prior works.

**Compliance With Llm Reviewing Policy:**

Affirmed.

**Key Questions For Authors:**

1. Can similar performance gains be expected when using other encoder models?

2. Would it also be possible to provide additional analysis for cases where the model itself is improperly learned (for example the biased model such as those studied in the Learning from Failure paper? In such cases, what kinds of phenomena would emerge?

**Limitations:**

The discussion of related work is somewhat limited. For example "Mitigating dataset bias by using per-sample gradient, ICLR'23" paper could also be included in the related work section.

In addition, I believe that incorporating the extra analyses mentioned above, particularly those raised in the questions and weaknesses, would further strengthen and improve the paper.

**Strengths And Weaknesses:**

## S1.
This paper presents an interesting problem setting by proposing a tool to analyze samples that are consistently misclassified by the model, offering a useful perspective for understanding systematic failure modes.

## S2.
The paper is well organized and clearly written, and the overall framework is presented in an intuitive and easy to follow manner.

## W1.
Although the analysis is meaningful, the paper would benefit from a clearer discussion of how the discovered slices can be used in practice, such as for debiasing, targeted retraining, or data collection strategies for hard examples.

## W2.
The method uses a pre-trained ResNet18 as the concept encoder, but the paper does not sufficiently analyze the potential issues arising from this choice. In particular, biases or limitations inherited from the encoder may propagate into conpcet prediction errors and affect the reliability of the identified slices.

---

> ### Author Rebuttal · Authors · 2026-03-30
>
> Thank you for your very valuable feedback. We are happy you found our paper “well organized and clearly written” and that it offers  “a useful perspective for understanding systematic failure modes”.
>
> Below, we address the questions and concerns raised in your review. If you have further concerns, please let us know. Otherwise, we would sincerely appreciate it if you would consider updating your score in light of these replies.
>
> ## (W1) How can discovered slices be used in practice?
>
> This is a great question, and it opens many avenues for future exploration. Once error slices have been identified and some of the problematic slices have been assigned semantics via the explanations generated by CB-SLICE (e.g., “Waterbirds on land backgrounds”), one can use that information as follows:
> 1. Error slices can inform potential future data collection. For example, one may decide to collect more images of waterbirds on land backgrounds once we know they are a minority group in our dataset.
> 2. If data collection is not possible, then error slices can still be used to inform potential data augmentation strategies during training. For example, colour-based spurious correlations identified from a slice detected by CB-SLICE can suggest introducing colour jittering during training as a strategy to improve model generalisation.
> 3. Moreover, data slices can themselves serve as subgroups for running group-supervised bias mitigation pipelines (e.g., GDRO [1]). For example, each slice could be treated as its own protected subgroup, and this can be used to train an unbiased model by minimising the worst-case group loss (as GDRO would do).
> 4. Finally, if data collection or retraining is not possible, then data slices can at the very least be used to update a model’s card [2] so that it reflects the model underperforms on certain subgroups, and that the model should be used with caution in setups where it may propagate or amplify these known biases.
>
> To take this feedback into account, we will update our discussion of future work in Section 7 of our manuscript to call for exploring these research venues.
>
> ## (Q1) Consequences of Encoder Choice
>
> Thank you for bringing up this concern. We reran experiments on the Waterbirds dataset using a Densenet-121 backbone for the encoder, rather than a ResNet-18 backbone. Our results, summarised in **[this table](https://anonymous.4open.science/r/anon-reb/encoder.pdf)**, show the same trends we observed with the ResNet18 encoder. This suggests that our results are not specific to the encoder we used for all baselines.
>
> To address this concern, we will update our manuscript to include these results in a new appendix, which will be briefly summarised in Sections 6.1 and 6.2.
>
>
> ## (W2 and Q2) Phenomena emerging when the model itself is improperly learnt (i.e., the encoder is biased)
>
> This question directly relates to how CB-SLICE works, precisely because CB-SLICE is designed to identify error slices even when the encoder itself is biased. To clarify why this is the case, consider a similar MNIST-Sum task as the one we used in our work but instead of colouring 90% of all $(1, 1)$ pairs with red, we colour red 90% of all pairs of the form $(1, x)$ (where $x \in [0, 1, 2, 3]$). Training a concept encoder will likely lead to a biased concept encoder that exploits colour spurious correlations when predicting the concept indicating whether the left digit is $1$ (i.e., $c_{\mathrm{left\ digit\ 1}}$).
>
> In this setup, a CBM trained with this encoder will likely be unable to accurately predict the **task label** for samples where $c_{\mathrm{left\ digit\ 1}} = 1$ but $c_\mathrm{red\ colour} = 0$ (since this is a minority class in the training set). But CB-SLICE is designed precisely to find such groups! Our clustering algorithm will identify that there is a slice of samples on the validation set where the downstream task is incorrect whenever $c_{\mathrm{left\ digit\ 1}}$ is mispredicted as $0$, and $c_\mathrm{red\ colour}$ is correctly predicted as $\hat{c}_\mathrm{red\ colour} = 0$. Therefore, because the concept encoder itself was biased (as a consequence of the training data), we can exploit this property to identify an error group that requires attention in the original training set.
>
> To take this feedback into account, we will update our discussion in Section 6.1 to describe how CB-SLICE can still detect error slices if the concept encoder itself is biased.
>
> ## (Limitation 1) Additional previous works
>
> Thank you so much for bringing “Mitigating dataset bias by using per-sample gradient” to our attention! We will make sure to discuss this paper in our updated manuscript. Specifically, we update our related work discussion to contrast this work with our approach.
>
> —
>
> ## References
>
> 1. Sagawa et al. "Distributionally robust neural networks for group shifts." ICML (2019).
> 2. Mitchell et al. "Model cards for model reporting." FAccT (2019).

---

> > ### Author Rebuttal · Reviewer_tXeA · 2026-04-04
> >
> > Thank you for the detailed response. I have carefully reviewed the authors' responses and will maintain my original score.

---

### Decision · Program_Chairs · 2026-04-30

**Decision:**

Accept (regular)

**Comment:**

This paper proposes a concept-based slice discovery method that leverages Concept Bottleneck Models (CBMs) to identify systematic error slices in deep learning models.

Reviewers are overall positive about this paper and acknowledge the contribution of exploiting CBMs for error slice discovery, a use case they have not previously been applied to (Reviewer GYQG); strong and consistent empirical performance across multiple benchmarks, systematically outperforming competing methods (Reviewer GYQG, Reviewer tEpV); and clear writing and well-organized presentation (Reviewer tXeA, Reviewer tEpV).

Following the rebuttal, all 3 reviewers confirmed their concerns were fully addressed, with Reviewer GYQG raising their score to Accept.
Therefore, the AC recommends accept for this paper.